# Mutations in the *KIF21B* kinesin gene cause neurodevelopmental disorders through imbalanced canonical motor activity

Laure Asselin[1,2,3,4], José Rivera Alvarez [1,2,3,4,20], Solveig Heide[5,6,7,20], Camille S. Bonnet[1,2,3,4,20], Peggy Tilly[1,2,3,4], Hélène Vitet [8], Chantal Weber[1,2,3,4], Carlos A. Bacino[9,10], Kristin Baranaño[11], Anna Chassevent[11], Amy Dameron[12], Laurence Faivre[13,14], Neil A. Hanchard [9], Sonal Mahida[15], Kirsty McWalter [12], Cyril Mignot[5,6,7,16], Caroline Nava[5,16], Agnès Rastetter[16], Haley Streff[9,10], Christel Thauvin-Robinet[13,17], Marjan M. Weiss [18], Gladys Zapata[10], Petra J. G. Zwijnenburg[18], Frédéric Saudou [8], Christel Depienne [1,2,3,4,16,19], Christelle Golzio[1,2,3,4], Delphine Héron[5,6,7] & Juliette D. Godin [1,2,3,4✉]

KIF21B is a kinesin protein that promotes intracellular transport and controls microtubule dynamics. We report three missense variants and one duplication in *KIF21B* in individuals with neurodevelopmental disorders associated with brain malformations, including corpus callosum agenesis (ACC) and microcephaly. We demonstrate, in vivo, that the expression of *KIF21B* missense variants specifically recapitulates patients' neurodevelopmental abnormalities, including microcephaly and reduced intra- and inter-hemispheric connectivity. We establish that missense *KIF21B* variants impede neuronal migration through attenuation of kinesin autoinhibition leading to aberrant KIF21B motility activity. We also show that the ACC-related *KIF21B* variant independently perturbs axonal growth and ipsilateral axon branching through two distinct mechanisms, both leading to deregulation of canonical kinesin motor activity. The duplication introduces a premature termination codon leading to nonsense-mediated mRNA decay. Although we demonstrate that *Kif21b* haploinsufficiency leads to an impaired neuronal positioning, the duplication variant might not be pathogenic. Altogether, our data indicate that impaired KIF21B autoregulation and function play a critical role in the pathogenicity of human neurodevelopmental disorder.

A list of author affiliations appears at the end of the paper.

The development of the mammalian cerebral cortex depends on microtubule (MT)-related processes that coordinate birth, migration and differentiation of excitatory and inhibitory neurons. MT cytoskeleton acts in concert with microtubule associated proteins (MAP) and motor proteins to promote the structural changes that underlie key developmental events such as neurogenesis, migration, neuritogenesis, axon pathfinding and synapse formation. Kinesin superfamily proteins (KIFs) are important molecular motors that control MT organization and dynamics in both axons and dendrites and mediate intracellular transport of various cargo, including vesicles, organelles, cellular proteins and mRNAs, along MTs[1,2]. The importance of both the force-generating and MT-regulating functions of KIFs for brain development has become evident with loss of function studies demonstrating defects in mitosis[3–9], cytokinesis[10,11], polarity[3], migration[12–14], axonal growth and branching[14–18], survival[19] and synaptogenesis[20–24]. Further reflecting the key role of KIFs in neuronal development, variants in human KIF-encoding genes (KIF4A[24], KIF7[25–28], KIF11[29], KIF2A[30–32], KIF5C[24,28,30,33], KIF1A[28,34], KIF6[35], KIF14[11,36,37], KIF26A[28]) have been associated with various neurodevelopmental disorders, including malformation of cortical development (MCD), acrocallosal syndrome, ciliopathies, epilepsy and intellectual disability. Most KIF variants have been predicted to be highly pathogenic in silico but their direct implication in disease and the underlying pathophysiological mechanisms have only been elicited for only a few of them[11,26,35,38].

The MT-plus-end directed kinesin-4 motor KIF21B is mainly expressed in spleen, testes and central nervous tissues and is particularly enriched in neurons[39,40]. Within neurons, KIF21B is present in both axons and dendrites and especially abundant in growth cones[40,41]. KIF21B has dual functions in neurons. First, it promotes intracellular transport through its N-terminal processive motor activity[21,42–44]. However, except for BDNF-TrkB signaling endosomes[44], 2-subunit-containing GABAA receptor[45] and neurobeachin recycling endosomes[43], our knowledge of KIF21B-transported cargoes is very limited. Second, KIF21B influences MT dynamics through distinct MT binding domains[21,42,44]. KIF21B positively regulates MTs dynamicity in dendrites by favoring MT growth and catastrophes[21,44]. In addition, in vitro studies demonstrate that KIF21B can also act as a MT pausing factor by accumulating at the MT-plus ends[42]. Notably, KIF21B functions can be modulated by neuronal activity, which favors KIF21B trafficking activity at the expense of MT dynamics regulatory function[44] as well as through an auto-inhibitory interaction between the N-terminal motor domain and an internal regulatory coiled-coil region (rCC)[42,46].

KIF21B homozygous knockout (KO) mice display severe morphological abnormalities including microcephaly and partial loss of commissural fibers[47], cognitive deficits[21,43,48] and altered synaptic transmission[21,23]. Together with the reduced dendritic complexity and spines density observed in KIF21B[−/−] neurons in culture[21], these results highlight a critical role for KIF21B in brain development and function. Though, there are no clear KIF21B-related neurodevelopmental disorders, a duplication of the locus bearing KIF21B has been found in individuals with neurodevelopmental delay and intellectual disability (ID)[49].

Here we provide the evidence of a causal relationship between variants in KIF21B and neurodevelopmental disorders. We report the identification of three missense variants and one truncating variant in patients with neurodevelopmental delay and brain malformations including corpus callosum (CC) agenesis (ACC) and microcephaly. By combining in vivo modeling tools, we show that KIF21B pathogenic variants impede neuronal migration and connectivity through at least two distinct mechanisms both leading to dysregulation of canonical kinesin motor activity.

Taken together our data suggest that KIF21B is a novel gene for ID associated with heterogeneous brain morphological anomalies.

## Results

**Identification of human KIF21B variants.** Using trio whole-exome sequencing, we identified a de novo variant (NM_001252100.1, c.2032A>C, p.Ile678Leu) in the KIF21B gene in a first patient (P1) presenting with developmental delay, learning and motor disabilities, associated with isolated complete agenesis of the corpus callosum (ACC) (Fig. 1a, e, Table 1, Supplementary Note 1). Through the GeneMatcher platform[50], variants in KIF21B were found in three additional patients. Patient 2 (P2) (NM_001252100.1, c.937C>A, p.Gln313Lys) presented with severe ID associated with microcephaly (Fig. 1b, f); patient 3 (P3) (NM_001252100.1, c.3001G>A, p. Ala1001Thr) presented with global developmental delay and mild to moderate ID (Fig. 1c) but normal brain structure at the MRI. This variant was inherited from the father, who presented with developmental delay and learning difficulties; and patient 4 (P4) (NM_001252100.1, c.2959_2962dup, p.Asn988Serfs*4) presented with mild developmental delays and hypotonia, but no brain structural anomalies on brain MRI (Fig. 1d, g). The three identified KIF21B missense variants occur within highly conserved residues positioned in the motor domain (NM_001252100.1, c.937C>A, p.Gln313Lys-P2), the regulatory coiled-coil (rCC) region (NM_001252100.1, c.3001G>A, p.Ala1001Thr-P3), and the coiled-coil domain (NM_001252100.1, c.2032A>C, p.Ile678Leu-P1) (Fig. 1h, Supplementary Fig. 1a–c). The fourth variant is a duplication (NM_001252100.1, c.2959_2962dup, p.Asn988Serfs*4-P4) that leads to the introduction of a premature termination codon in exon 20. RT-qPCR analysis and sequencing of KIF21B transcripts isolated from P4's blood revealed haploinsufficiency, likely due to the degradation of the mutant mRNA by nonsense-mediated decay (Supplementary Fig. 1d, e). All variants were predicted pathogenic by commonly used in silico software (Polyphen-2, Mutation Taster, SIFT and CADD; Supplementary Fig. 1f) and co-segregated with the phenotype in each pedigree (Fig. 1a–d). Of note, we found two other de novo variants of unknown significance in patients: one hemizygous variant in ARHGAP4 (chromosome X) in P1 that also segregated in his healthy brother and one de novo in UBR3 (NM_172070.3, c.5023G>C; p.Glu1675Gln) in P4, that showed a weak pathogenic score based on in silico predictions. None of the four KIF21B variants is reported in public databases, including dbSNP, 1000 Genomes and gnomAD. Overall, we identified variants in KIF21B gene in four patients presenting with mild to severe neurodevelopmental delay associated with heterogeneous brain malformations (Table 1, Supplementary Note 1).

**Kif21b expression is restricted to neurons.** We first examined the expression pattern of Kif21b in the mouse developing cortex. Although levels of Kif21b mRNA transcripts are rather stable during development (Fig. 2a), protein expression tends to increase from embryonic day (E) 12.5 to postnatal day (P) 2 (Fig. 2b). mKif21b transcripts were mostly observed in the cortical plate (CP) and almost entirely excluded from the ventricular (VZ) and subventricular (SVZ) zones, where progenitors and newborn neurons reside (Supplementary Fig. 2a). As corticogenesis proceeds, mKif21b mRNAs accumulated in the intermediate zone, which is enriched in growing axons (Supplementary Fig. 2a). Immunolabeling of E12.5 to E18.5 embryo brain sections showed a restricted expression of mKif21b proteins to postmitotic compartments of the neuronal epithelium with a particularly intense signal in the axon-rich zone

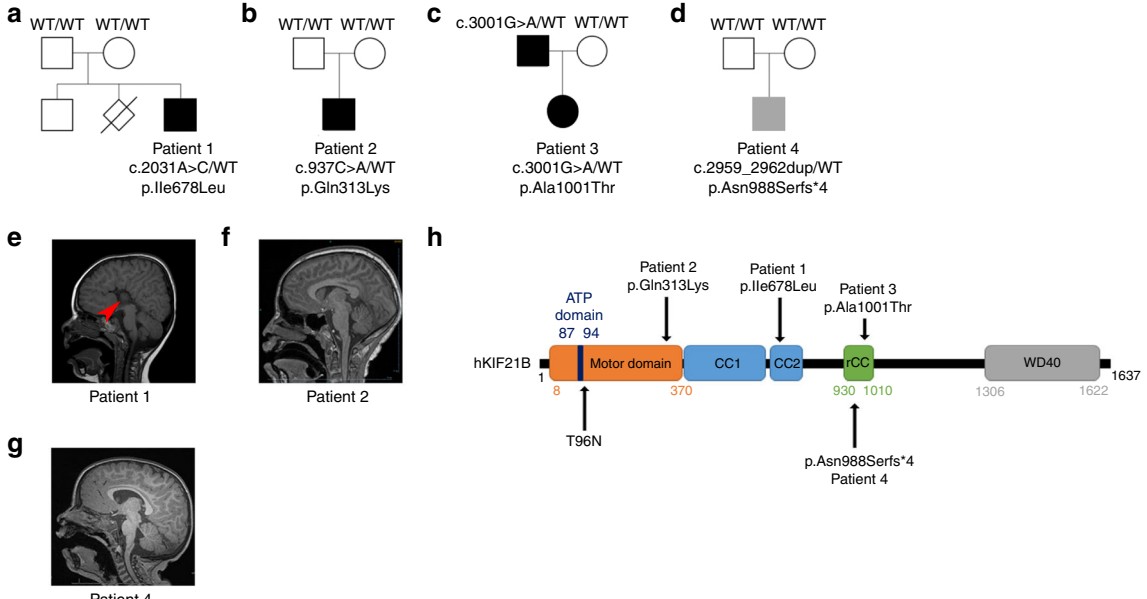

**Fig. 1 Patients with *KIF21B* variants. a–d** Pedigrees of patients with identified *KIF21B* variants. **e–g** Sagittal brain section of patient's MRI showing a complete agenesis of the corpus callosum in patient 1 (**e**, red arrow) and microcephaly in patient 2 (**f**). **h** Schematic representation of the human KIF21B (hKIF21B) protein indicating the different domains (motor domain, ATP binding site, coiled-coil domain (CC) 1 and 2, regulatory coiled-coil domain (rCC) and WD40 domain) and the position of the mutated amino acids for patient 1 (p.Ile678Leu), 2 (p.Gln313Lys), 3 (p.Ala1001Thr) and 4 (p.Asn988SerfsX4). T96N substitution that abolishes KIF21B mobility is also depicted.

(Fig. 2c and Supplementary Fig. 2b, c). mKif21b staining specificity was validated by the absence of labeling in *Kif21b* knockout brain sections at birth (Supplementary Fig. 2e, f). Further immunostainings with antibodies against mKif21b and βIII-Tubulin or Tbr2 that label neurons and intermediate progenitors respectively, confirmed that Kif21b is exclusively expressed in post mitotic neurons at all stages of development (Fig. 2c, d and Supplementary Fig. 2b, c). To undoubtedly exclude any expression of Kif21b in progenitors, we analyzed mKif21b protein in progenitor (YFP⁻; CD24⁻) and neuron populations (YFP⁺; CD24⁺) isolated from Rosa26-lox-STOP-YFP; NEX^CRE/⁺ E16.5 cortices by fluorescent-activated cell sorting. No Kif21b protein was detected in cortical progenitors by immunoblotting (Supplementary Fig. 2d). Although Kif21b was previously thought to be mainly localized to dendrites[40], our data suggest an axonal localization in the developing cortex (Supplementary Fig. 2b). To ascertain the axonal distribution of mKif21b, we used a 2-chambers microfluidic device to analyze separately the axons from cell bodies and dendrites[51] (Fig. 2e, f). Channels separating both chambers are 450-μm-long so only axons from neurons located in the proximal compartment can reach the distal chamber[51]. Immunolabelings of mouse primary cortical neurons with Kif21b and Tau antibodies in these devices confirmed the expression of Kif21b in Tau-positive axons and its accumulation at the axonal tips (Fig. 2e, f). Kif21b is thus moderately but stably expressed in both dendrites and axons of postmitotic neurons from early to late corticogenesis.

**KIF21B variants impair migration of projection neurons**. To evaluate the pathogenic nature of *KIF21B* variants, we assessed the consequences of overexpressing h*KIF21B* variants on neuronal migration using in utero electroporation (IUE) in mice. Given that KIF21B is a post mitotic kinesin, we used plasmids allowing expression of human KIF21B under the control of the NeuroD promoter (NeuroD-hKIF21B). Transfection in N2A neuroblastoma cell line showed even expression of wild-type (WT) and all three missense hKIF21B variants by western blot, suggesting that

p.Gln313Lys, p.Ile678Leu and p.Ala1001Thr missense variants are unlikely to affect the production, stability or turnover of the hKIF21B proteins (Supplementary Fig. 3a). Consistently, cycloheximide chase experiments in N2A cells revealed a similar half-life of WT and mutant hKIF21B proteins (Supplementary Fig. 3b).

To investigate the effects of the variants on neuronal positioning, we individually induced neuron-specific expression of the hKIF21B mutants using IUE of NeuroD-hKIF21B constructs together with a NeuroD-IRES-GFP reporter plasmid in mouse embryonic cortices at E14.5. Four days after IUE, whereas most of the GFP+ postmitotic neurons expressing full-length WT-hKIF21B reached the CP as in the control (Fig. 3a), neurons expressing missense variants accumulated in the intermediate zone, with a decrease of 27.7%, 60.3% and 23% of the cells reaching the upper CP in the p.Gln313Lys, p.Ile678Leu and p.Ala1001Thr conditions, respectively (Bonferroni adjusted $P = 0.0001$) (Fig. 3b). Noteworthy, h*KIF21B* missense variants likely disturbed neuronal migration in a cell-autonomous manner as their expression did not affect cell survival and glia scaffold integrity (Supplementary Fig. 3c). To assess the functional consequences of the p.Asn988Serfs*4 protein truncated variant (Supplementary Fig. 1d), we silenced m*Kif21b* specifically in post mitotic neurons using IUE of CRE-dependent inducible shRNA vector[52] together with a NeuroD-CRE-IRES-GFP construct at E14.5. Efficacy of the two shRNAs was confirmed by RT-qPCR (−61.4% for sh-*Kif21b* #1, −45.1% for sh-*Kif21b* #2) (Supplementary Fig. 3d). Four days after IUE, *Kif21b*-silenced neurons displayed migration defects compared to control shRNA-electroporated cells with a reduction of 23.5% and 32.2% of cells distributed in the upper CP for sh-*Kif21b* #1 and sh-*Kif21b* #2, respectively (Supplementary Fig. 3e, f). To note, the migratory phenotype induced by sh-*Kif21b* #2 was fully recovered by co-electroporation of wild-type NeuroD-mKif21b construct (Supplementary Fig. 3e, f). Most of the cells overexpressing the p.Gln313Lys and p.Ala1001Thr mutants or silenced for *Kif21b* showed a correct positioning with nearly all cells found in

**Table 1 Clinical summary of patients with *hKIF21B* variants.**

| | Patient 1 | Patient 2 | Patient 3 | Patient 4 |
|---|---|---|---|---|
| Age at last evaluation | 10 y | 12 y 1 m | 9 y | 3 y 8 m |
| Sex | Male | Male | Female | Male |
| **Genetics** | | | | |
| Gene | KIF21B | KIF21B | KIF21B | KIF21B |
| NM_001252100.1 HGVS protein nomenclature | c.2032A>C p.Ile678Leu | c.937C>A p.Gln313Lys | c.3001G>A p.Ala1001Thr | c.2959_2962dupGCCA p.Asn988Serfs*4 |
| Inheritance | De novo | De novo | Inherited from the affected father | De novo |
| **Pregnancy and delivery** | | | | |
| Pregnancy | Normal | Oligohydramnios and IUGR | Normal | Small for gestational age |
| Height (perc)/weight (perc)/head circumference (perc) at birth | 51 cm (97th p), 3.750 kg (97th p), 33 cm (15th p) | 49 cm (49th p), 2.584 kg (7th p), 32 cm (5th p) | 3.480 kg (50th p) | 46.4 cm (7th p), 2.633 kg (5.87th p), *34 cm (25th p) *Measurements done at age 6 days |
| Neonatal findings | None | Nuchal cord at birth and was blue, mild respiratory distress, but discharged with mother | None | Feeding issues, NG tube |
| **Developmental stages** | | | | |
| Age of sitting (months) | 10 | Does not | 14 | 9 |
| Age of walking (months) | 18 | Does not | 24 | 15–16 |
| Language delay | Yes | Yes | Yes | No |
| Age of first words (m: months, y:years) | 36 m | | 36 m | 12 m |
| Age of first sentences (m:months, y:years) | NA | | N/A | 24 m |
| Current language ability | Short sentences, dysarthria | Non-verbal | Sentences | Short sentences |
| **Intellectual disability (ID)** | | | | |
| Estimated level of ID (mild, moderate, severe) | Borderline | Severe | Mild to moderate | Mild |
| Age at evaluation (y) | 10 y | | 9 y | |
| Total IQ | 66–79 (WISC V) | | 54–59 (WISC V) | |
| **Clinical examination** | | | | |
| Age at examination (years) | 5 y | 12 y 1 m | 3 y 9 m | 3 y 8 m |
| Height (SD)/weight (SD)/head circumference (SD) | 118 cm (+1.5), 21.3 kg (+2), 52 cm (+0.5) | 139 cm (−1.3), 20 kg (−3.4), 48.5 cm (−3.9) | 99 cm (−1.1), 17.8 kg (+1,8), 49.6 cm (−0.2) | 90.7 cm (−1.67), 16.3 kg (+0.09), 50.7 cm (+0.72) |
| Neurologic examination | Slow | Poor visual fixation, constant tongue thrusting, poor gag, poor head control, bilateral ankle tightness, right wrist contracture | Mildly hypertonic legs | Hypotonia |
| Dysmorphic features | Plagiocephaly | Large eyes, fleshy ears, hypertelorism | Epicanthic folds, mild ptosis, tented upperlip | Upslanting palpebral fissures, prominent eyebrows, broad nose with bulbous tip, anteverted nares. Micrognathia. Right-sided Duane syndrome |
| **Brain imaging (MRI)** | | | | |
| Age at examination (m: months, y:years) | 3 y | 6 m and 12 y | 2 y | 1 y 9 m |
| Brain anomalies (MRI) | Complete agenesis of the corpus callosum | Normal | No structural abnormalites, myelinisation not completed yet. Normal differentiation white and gray matter. No focal lesions. Normal spectroscopy | A few scattered punctate foci of T2 prolongation in subcortical white matter and periventricular white matter of bilateral cerebral hemispheres with no associated restricted diffusion or hemorrhage |
| Other | | | | History of falling spells with normal EEG. Severe constipation, Central sleep apnea, History of feeding issues requiring G tube |

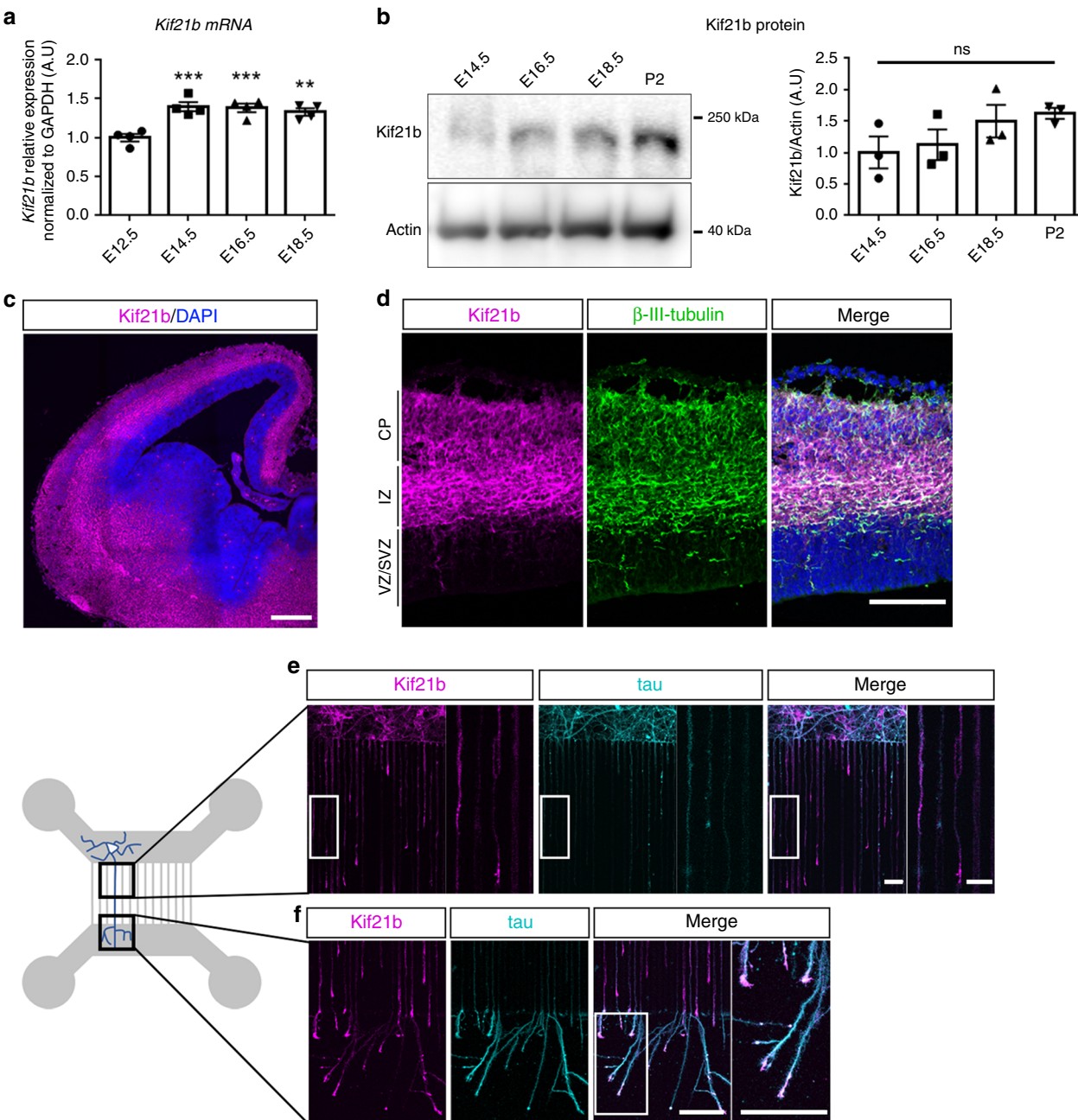

**Fig. 2 Kif21b expression in mouse developing cortex. a** RT-qPCR analyses showing expression of *Kif21b* transcripts in mouse cortices at different embryonic stages (from E12.5 to E18.5) (*n* = 4 brains per stage). **b** Western blot analyses of mouse cortical extracts showing similar expression of Kif21b protein from E14.5 to P2 (*n* = 3 brains per stage). **a**, **b** Data are represented as means ± S.E.M. Significance was calculated by one-way ANOVA (Bonferroni's multiple comparisons test), ns non-significant; **\*\***P < 0.005; **\*\*\***P < 0.001. **c**, **d** E14.5 mouse forebrain coronal sections immunolabelled for Kif21b (magenta) and β-III-tubulin (neuronal marker, green) and counterstained with DAPI (blue) showing that Kif21b expression is restricted to post mitotic neurons. **e**, **f** Left panel: schematic representation of a 2-compartments microfluidic chamber. Cortical neurons (in cyan) plated in the upper chamber (gray) grow their axons through 450-μm-long microchannels. The length of the microchannels allows axons but not dendrites to reach the lower chamber. Right panel: immunolabeling of Kif21b (magenta) and tau (axonal marker, cyan) on mouse primary cortical neurons at DIV5 in microdevices showing expression of Kif21b in axons (**e**) with an enrichment in growth cones (**f**). CP cortical plate, IZ intermediate zone, SVZ subventricular zone, VZ ventricular zone. Scale bars, (**c**) 250 μm and (**d**–**f**) 100 μm, magnifications (**e**, **f**) 20 μm. Source data are provided in the Source Data file.

the upper layer of the cortex after birth, indicating a delay in migration rather than a permanent arrest (Fig. 3c, d, Supplementary Fig. 3g, h). By contrast, the p.Ile678Leu variant induced a permanent migration defect as a large number of p. Ile678Leu-expresing neurons remained in the white matter and deep-layers at P2 (Fig. 3c, d). Remarkably, p.Ile678Leu-expressing

projection neurons permanently arrested in the white matter were expressing the upper-layer marker Cux1, which supports a faulty migration rather than specification defects (Fig. 3e). Altogether, these results demonstrate that missense h*KIF21B* variants and *Kif21b* haploinsufficiency impede, to various extents, the radial migration of projection neurons.

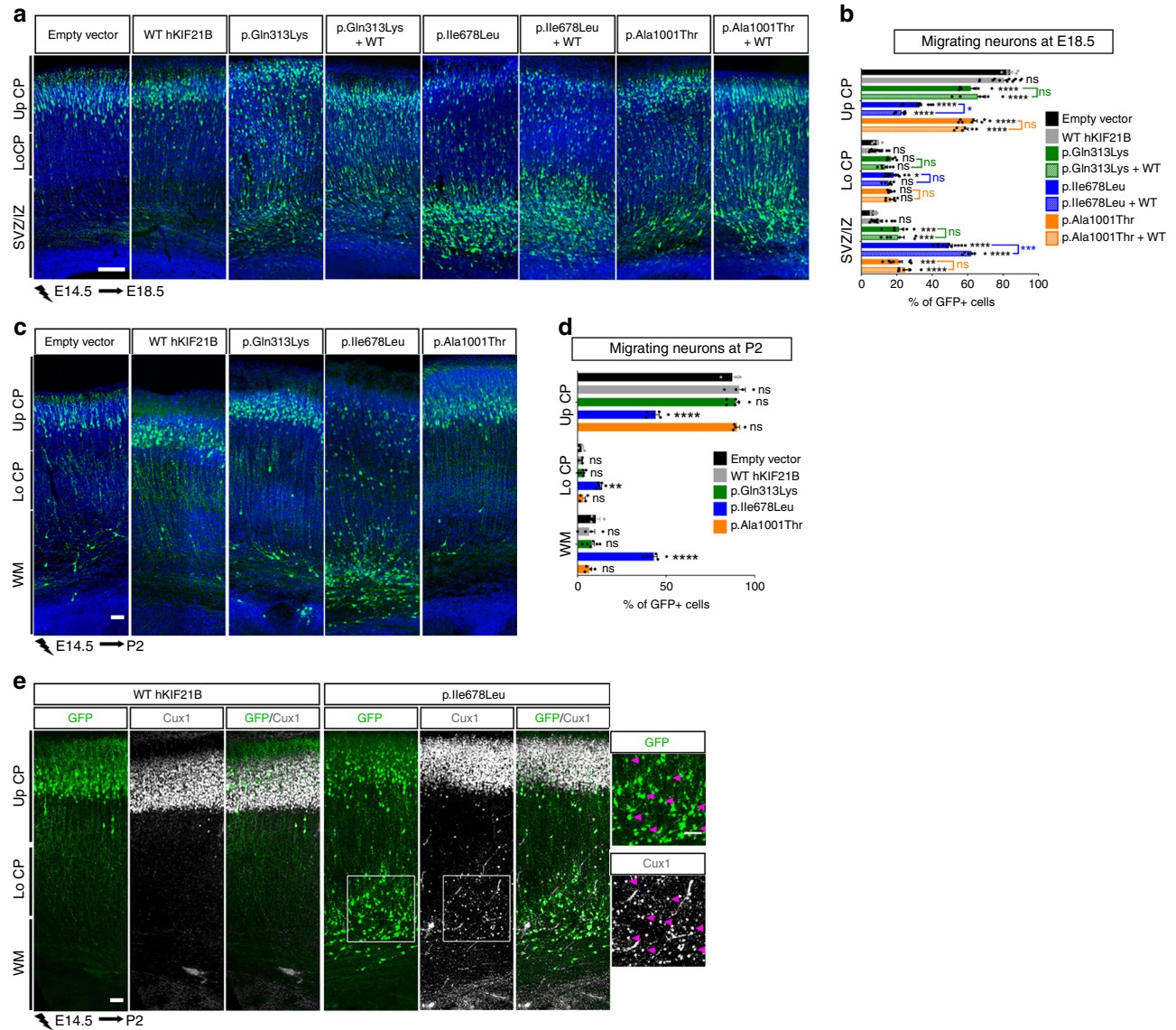

**Fig. 3 Expression of h*KIF21B* missense variants induces abnormal neuronal migration. a, c** Coronal sections of E18.5 (**a**) or P2 (**c**) mouse cortices electroporated at E14.5 with NeuroD-IRES-GFP empty vector (1 μg/μL) or WT, p.Gln313Lys, p.Ile678Leu or p.Ala1001Thr NeuroD-hKIF21B or co-expressing WT-hKIF21B together with mutated hKIF21B constructs (ratio 1:1). GFP-positive electroporated cells are depicted in green. Nuclei are stained with DAPI. **b, d** Analysis (means ± S.E.M.) of the percentage of electroporated GFP-cells in different regions (Up CP: upper cortical plate, Lo CP: lower cortical plate, IZ: intermediate zone, SVZ: subventricular zone) showing effect of expressing hKIF21B variants. Data were analyzed by two-way ANOVA (Bonferroni's multiple comparisons test). Number of embryos analyzed: **b** Empty vector, $n = 6$; WT, $n = 13$; p.Gln313Lys, $n = 6$; p.Ile678Leu, $n = 11$; p.Ala1001Thr, $n = 8$. Rescue experiments: $n = 6$ for each condition. **d** Empty vector, $n = 3$; WT, $n = 4$; p.Gln313Lys, $n = 6$; p.Ile678Leu, $n = 6$; p.Ala1001Thr, $n = 4$. ns non-significant; *$P < 0.05$; **$P < 0.005$; ***$P < 0.001$; ****$P < 0.0001$. **e** Cux1-immunolabeling (gray) of P2 coronal sections of mouse brains electroporated at E14.5 with the WT or p.Ile678Leu NeuroD-hKIF21B constructs showing no specification defects of arrested neurons (green). Scale bars (**a, c, e**) 50 μm. Source data are provided in the Source Data file.

***KIF21B* variants lead to aberrant KIF21B motility activity**. To understand the molecular mechanisms by which variants in h*KIF21B* gene lead to defective radial migration, we tested for restoration of the h*KIF21B* variant-induced phenotype by increasing amount of wild-type protein. IUE of wild-type hKIF21B together with hKIF21B mutants at a 1:1 ratio failed to rescue the migration phenotype (Fig. 3a, b). Strikingly, neurons overexpressing large amount of WT-hKIF21B (2 units of NeuroD-hKIF21B) failed to reach the upper CP 4 days after IUE (Fig. 4a, b). Collectively, these results raise the possibility that h*KIF21B* variants impair migration by enhancing KIF21B activity in a dominant manner. One possible mechanism by which hKIF21B mutants might exert this effect is by relieving autoinhibition imposed by the rCC to the motor domain as shown for CFEOM-causing variants in KIF21A[46,53,54], a kinesin-4 family member that shares 61% identity with KIF21B[40]. Consistent with the hypothesis that KIF21B hyperactivation cause migration phenotypes, expression of a truncated mouse mKif21b protein that lacks the rCC domain (1 unit of NeuroD-mKif21bΔrCC) led to faulty migration (Fig. 4c, d).

To test whether identified variants alter KIF21B autoinhibition, we next sought to explore the functions of KIF21B that were enhanced by KIF21B autoinhibition release in mutant conditions. Given the processive activity of KIF21B, we assessed the effect of the variants on KIF21B motility activity. Using immunofluorescence, we first compared the localization of the wild-type (WT)

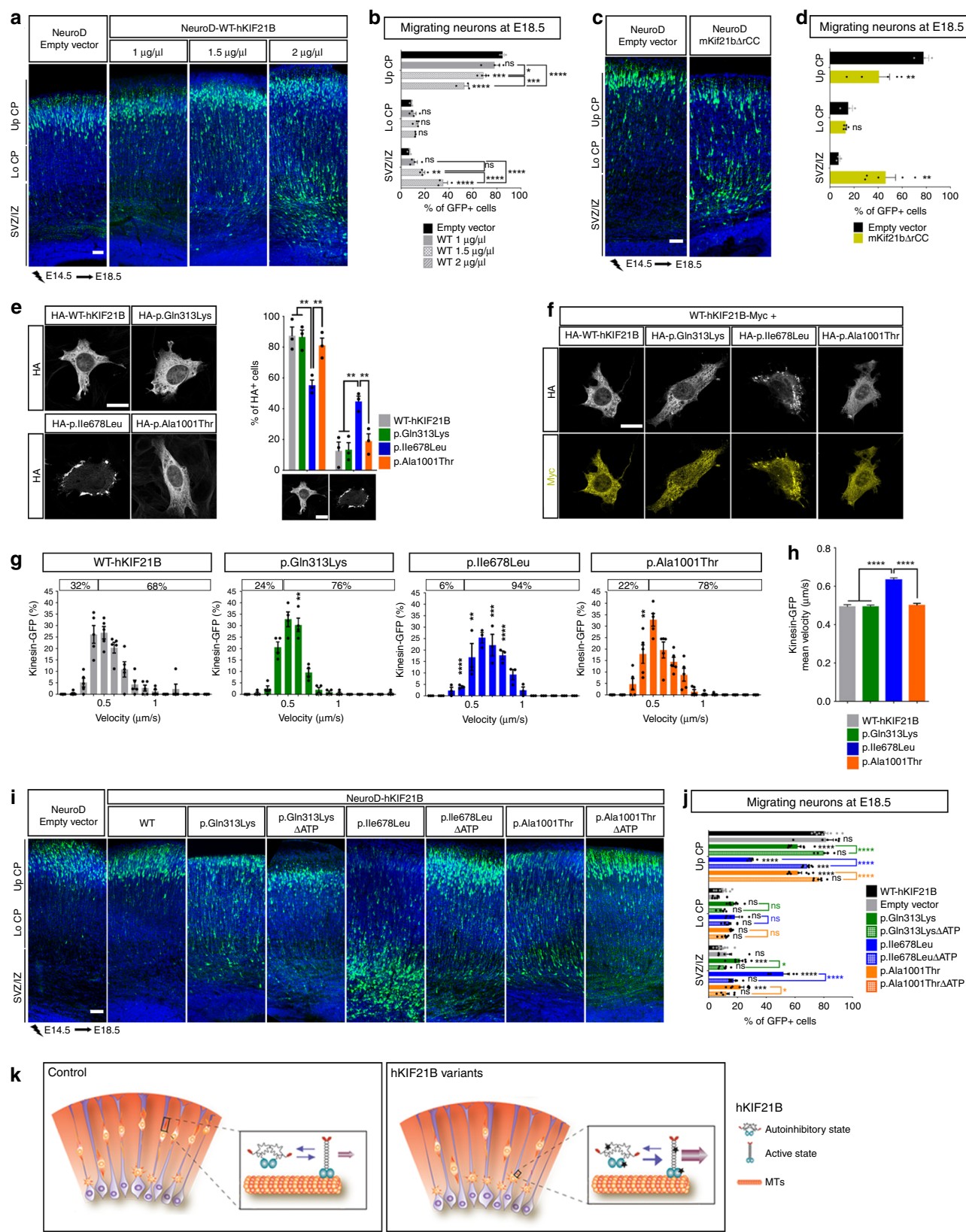

and the three missense mutant proteins in ST cells transfected with pcDNA-HA-hKIF21B cDNA constructs (Fig. 4e, Supplementary Fig. 4a). Although WT, p.Gln313Lys and p.Ala1001Thr proteins showed similar diffuse cytoplasmic localization, the p. Ile678Leu KIF21B protein tended to form aggregates localized mainly at the periphery of the cells, suggesting an enhanced motility toward the plus end of the microtubules (Fig. 4e). To note, WT and all mutant proteins showed similar distribution in soma and neurites when overexpressed in primary cortical neurons (Supplementary Fig. 4c). Interestingly, in 100% of the

**Fig. 4 h*KIF21B* variants induce abnormal migration through enhanced KIF21B motor activity. a, c, i** Coronal sections of E18.5 cortices, 4 days after IUE with the indicated NeuroD-IRES-GFP constructs. GFP-positive electroporated cells are depicted in green. Nuclei are stained with DAPI. **b, d, j** Percentage (means ± S.E.M.) of electroporated cells in upper (Up CP) and lower (Lo CP) cortical plate, intermediate (IZ) and subventricular zone (SVZ), showing the effect of increasing (**b**) amount or (**d**) activity of hKIF21B and (**j**) contribution of the processive activity to the phenotype. Data were analyzed by two-way ANOVA (Bonferroni's multiple comparisons test). Number of embryos analyzed: **b** Empty vector and WT 2 μg/μL, $n = 3$; WT 1 and 1.5 μg/μL, $n = 4$; **d** $n = 3$ for each condition; **j** Empty vector, $n = 9$; WT, $n = 6$; p.Gln313Lys, $n = 8$; p.Ile678Leu, $n = 5$; p.Ala1001Thr, $n = 8$; p.Gln313LysΔATP, $n = 5$; p.Ile678LeuΔATP, $n = 8$; p.Ala1001ThrΔATP, $n = 6$. **e** Immunolabeling of ST cells transfected with indicated HA-tagged hKIF21B constructs showing impaired localization of p.Ile678Leu variant. Histogram (means ± S.E.M.) represents the percentage of cells with a diffuse versus an impaired localization of the HA-tagged proteins. Data were analyzed by two-way ANOVA (Bonferroni's multiple comparisons test). Number of cells analyzed: WT, $n = 467$; p.Gln313Lys, $n = 429$; p.Ile678Leu, $n = 540$; p.Ala1001Thr, $n = 387$; from three independent experiments. **f** Immunolabeling of ST cells transfected with the indicated Myc- and HA-tagged hKIF21B constructs showing that the p.Ile678Leu variant alters the localization of the WT protein. **g, h** Live imaging of Cos7 cells transfected with the indicated GFP-tagged hKIF21B constructs. Histograms (means ± S.E.M.) represent (**g**) velocities distribution and (**h**) mean velocities. 20–32 cells from 3–5 independent experiments were analyzed by (**g**) two-way ANOVA or **h** one-way ANOVA (Bonferroni's multiple comparisons test). Total number of particles analyzed: WT, $n = 203$; p.Gln313Lys, $n = 182$; p.Ile678Leu, $n = 195$; p.Ala1001Thr, $n = 237$. ns non-significant; *$P < 0.05$; **$P < 0.005$; ***$P < 0.001$; ****$P < 0.0001$. Scale bars: (**a, c, i**) 50 μm; (**e, f**) 20 μm. **k** Model: WT KIF21B switches between an autoinhibition and an active (purple arrows) state. h*KIF21B* variants (marked by a star) conformation favors the active state. KIF21B hyper-motility (pink arrow) leads to migration defects. Source data are provided in the Source Data file.

ST cells where the p.Ile678Leu variant is mislocalized, the cellular localization of the WT protein is altered, suggesting that the p.Ile678Leu protein might act as a dominant negative protein (Fig. 4f). In accordance, the p.Ile678Leu variant is competing with the WT protein to form KIF21B homodimer. Indeed, anti-Myc immunoprecipitation on extracts from HEK293T cells expressing myc-tagged WT, HA-tagged WT and WT or p.Ile678Leu GFP-tagged hKIF21B proteins revealed that the binding of the Myc and HA-tagged WT proteins was affected by the expression of the p.Ile678Leu missense variant (Supplementary Fig. 4d). We further analyzed the processivity of mutant hKIF21B in Cos7 cells transfected with GFP-tagged hKIF21B constructs (Supplementary Fig. 4b). Live-cell imaging revealed a shift of GFP-hKIF21B velocity toward high speed for all variant proteins compared to the WT protein (Fig. 4g). Notably the p.Ile678Leu variant showed a more drastic effect with an increased average velocity of 28% compared to the WT protein (Fig. 4h). We next assessed the effect of h*KIF21B* variants on the trafficking of BDNF vesicles and mitochondria, two potential cargoes of KIF21B[21,44]. Fast videomicroscopy experiments performed in Cos7 cells transfected with WT and mutant pcDNA-HA-hKIF21B cDNA (Supplementary Fig. 4b) and BDNF-mCherry or Mito-RFP constructs did not reveal any change in the dynamics of neither BDNF-mCherry-containing vesicles (Supplementary Fig. 4e–g) or mitochondria (Supplementary Fig. 4h, i), suggesting that hKIF21B variants might lead to excessive motility of other unidentified cargoes. Collectively, these data indicate that the variants enhanced KIF21B processive activity through lessening of the kinesin autoinhibition.

We finally tested the effect of expressing immotile hKIF21B mutant proteins on neuronal migration. We performed IUE of truncated WT and mutant hKIF21B that lacks the ATP binding domain (Fig. 1h) in wild-type E14.5 mouse cortices. Although a significant number of neurons expressing the p.Gln313Lys, p.Ile678Leu and p.Ala1001Thr hKIF21B variants were trapped in the IZ at E18.5, most of the cells expressing the immotile variants (NeuroD-p.Gln313Lys-ΔATPhKIF21B, NeuroD-p.Ile678Leu-ΔAT-PhKIF21B, NeuroD-p.Ala1001Thr-ΔATPhKIF21B) showed a correct distribution (Fig. 4i, j, Supplementary Fig. 3a), demonstrating that preventing the motility of the mutant proteins decreases the severity of the migration phenotype. These results confirmed that the mutant protein impairs radial migration at least by enhancing KIF21B motility activity through the release of the kinesin autoinhibition (Fig. 4k). Collectively, our data indicate that modulation of kinesin autoregulation is critical in KIF21B-associated cortical migration phenotypes.

**KIF21B p.Gln313Lys variant reduced head size in zebrafish.** Considering the presence of microcephaly in the subject with p.Gln313Lys variant, we asked whether this mutant could induce head size defects in an appropriate animal model. We therefore turned toward the developing zebrafish embryo, a model that has been extensively used for microcephaly modeling, the measure of head size being a relevant proxy for brain size[55]. drKIF21B protein is broadly distributed in zebrafish larval brain at 5 days post-fertilization (dpf), a stage characterized by strong upregulation of *drKIF21B* transcripts (Supplementary Fig. 5a, b). Larvae injected with p.Gln313Lys human mRNA showed a significant and physiologically relevant reduction of head size compared to control at 5 dpf ($-6\%$, Welch two sample $t$-test $P = 1.185 \times 10^{-9}$), therefore exhibiting a phenotype analogous to the microcephaly observed in the human clinical condition (Fig. 5). By contrast, introduction of WT mRNAs or the two other missense variants (p.Ala1001Thr and p.Ile678Leu), that do not lead to head circumference defects in patients, barely affected zebrafish head size (Fig. 5, Supplementary Fig. 5c, d). These data suggest that this particular p.Gln313Lys variant in the motor domain of KIF21B likely drives the microcephaly phenotype observed in the individual carrier.

**p.Gln313Lys KIF21B variant does not impair proliferation.** We next sought to understand the mechanisms by which the p.Gln313Lys variant impairs brain size. Expression of h*KIF21B* missense variants in mouse cortical neurons did not induce cell death (Supplementary Fig. 3c), excluding the possibility that the microcephaly phenotype arises from a poor survival of neurons. Although KIF21B expression is restricted to neurons, we further tested whether expression of the p.Gln313Lys variant could non-cell autonomously affect the progenitors' biology. We performed IUE of NeuroD-p.Gln313Lys hKIF21B at E14.5 in wild-type cortices and analyzed the progenitors located in the electro-porated area at E16.5. The total number and the proliferative fraction (Ki67[+]) of both apical (Pax6[+]) and intermediate progenitors (Tbr2[+]) were indistinguishable from control condition (Supplementary Fig. 5e–j), suggesting that the brain size phenotype induced by the p.Gln313Lys hKIF21B variant was unlikely to have arisen from impaired neurogenesis. Altogether these results suggested that neither impaired birth nor poor survival of neurons or their progenitors contributed to the microcephaly phenotype observed in the subject with the p.Gln313Lys variant.

**p.Ile678Leu h*KIF21B* variant expression impedes axonogenesis.** The corpus callosum (CC), the major commissure connecting

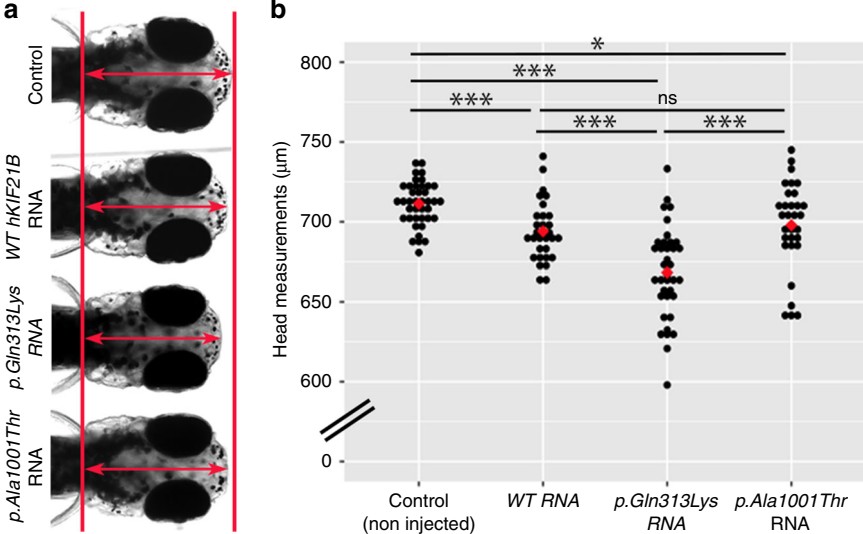

**Fig. 5 Expression of p.Gln313Lys hKIF21B decreases head size in zebrafish larvae. a** Dorsal view of representative control zebrafish larvae (non-injected) or injected with 100 pg of wild-type (WT) or mutated *hKIF21B* mRNAs (p.Gln313Lys and p.Ala1001Thr) at 5 days post-fertilization (5 dpf). Double arrow indicates the distance between the forebrain and hindbrain, a measure used as a proxy for head size. **b** Dot plot of the head measurements (red double arrow) of control and injected larvae at 5 dpf. Red diamond corresponds to the mean of the batch measured. Number of embryos analyzed for this specific batch: control, $n = 40$; WT, $n = 31$; p.Gln313Lys, $n = 38$; p.Ala1001Thr, $n = 31$. Experiments were repeated six times for non-injected control embryos ($n = 231$), four times for WT-injected embryos ($n = 127$), three times for p.Gln313Lys-injected embryos ($n = 108$) and four times for p.Ala1001Thr-injected embryos ($n = 158$). ns, non-significant, $*P < 0.05$, $**P < 0.005$, $***P < 0.001$. Significance was calculated by unpaired two-tailed Student's *t*-test or a Welch's two sample *t*-test between control and RNA-injected larvae. ns non-significant, $*P < 0.05$, $**P < 0.005$, $***P < 0.001$. Source data are provided in the Source Data file.

the two cerebral hemispheres, is formed of hundreds of millions of axons projecting contralaterally from the callosal projection neurons. Callosal axons cross the midline around birth to reach, in the first postnatal week, the contralateral cortex where they branch extensively at layer II/III and V. Given the ACC in the patient carrying the p.Ile678Leu variant, we investigated how the missense variants in h*KIF21B* lead to aberrant inter-hemispheric connectivity by introducing WT or mutant cDNA (NeuroD-hKIF21B) together with a mScarlet-expressing vector (pCAG2-mScarlet) in wild-type callosal projection neuron via IUE of E15.5 mouse cortical progenitors. The p.Gln313Lys substitution variant was used as a negative control as we did not expect any commissural defects according to the patient clinical features (Table 1). At P4, soon after the axons cross the midline and at P8, when axons start invading the contralateral CP, neither the expression of WT-hKIF21B nor of any of the variants perturbed midline crossing, as indicated by an equivalent scarlet intensity on each side of the CC (Fig. 6a–c, Supplementary Fig. 6a–c). In addition, at P22, when callosal axons achieve their adult-like arborization pattern, axons correctly invaded the homotopic contralateral cortex and successfully branched in layer II-III and V in all conditions (Supplementary Fig. 6g, h). Nonetheless, expression of the p.Ile678Leu, but not WT nor p.Gln313Lys mutant reduced by half the density of scarlet-positive axons in the white matter compared to the control both at P4 and P8 (−49.7% at P8, $P = 0.0043$) (Fig. 6a, b, d, e, Supplementary Fig. 6a, b, d). These defects were unlikely due to delayed innervation, as the poor inter-hemispheric connections persisted at P22 (Supplementary Fig. 6e, f). Rerouting through alternate commissures was also excluded as no aberrant axonal projections were observed after electroporation of callosal neurons and as no other commissures were shown enlarged in the patient (P1). Altogether, these results indicate that the faulty CC innervation is due to an impaired axonal growth rather than to defective contralateral targeting. Accordingly, we

measured the length of the longest neurites in primary cortical neurons transfected with pcDNA-HA-hKIF21B cDNA constructs. Although primary cortical neurons expressing WT-hKIF21B or p.Ala1001Thr and p.Gln313Lys hKIF21B variants showed normal axonal growth, expression of p.Ile678Leu mutant severely impaired axonogenesis at DIV2 and DIV5 (Fig. 6f, g, Supplementary Fig. 7a, b). To dig deeper into the pathogenic mechanism of the p.Ile678Leu hKIF21B variant, we performed a complementation experiment by co-electroporating p.Ile678Leu hKIF21B with increasing amounts of WT-hKIF21B at E15.5 and analyzed the percentage of projecting neurons at P8. Equivalent amounts of WT-hKIF21B failed to rescue the CC innervation phenotype (Fig. 6b, e). Further in vitro analysis of neurite length in primary cortical neurons expressing WT-hKIF21B together with p.Ile678Leu hKIF21B mutant at a 1:1 ratio confirmed the lack of rescue of axonal growth (Fig. 6f, g). We next reasoned that the p.Ile678Leu variant might exert its effect through attenuation of KIF21B autoinhibition. Accordingly, primary neurons transfected with two units of NeuroD-hKIF21B displayed shorter longest neurites in vitro (Fig. 6f, g), suggesting that enhanced KIF21B activity induces axonogenesis defects. To corroborate these findings in vivo, we assessed the ability of neurons electroporated with ΔATP p.Ile678Leu variant (NeuroD-p.Ile678Leu-ΔATPhKIF21B) to project axons contralaterally at P8 (Fig. 6b, c, e). There was no significant difference between the control and the immotile p.Ile678Leu-ΔATPhKIF21B, suggesting that the p.Ile678Leu variant impedes CC innervation through aberrant motor activity. Consistent with a hyperactivation of KIF21B, neurons expressing NeuroD-mKif21bΔrCC failed to project axons at P8 (Fig. 6b, e). Altogether these results indicate that the loss of inter-hemispheric connectivity induced by the p.Ile678Leu hKIF21B variant, and the subsequent release of KIF21B autoinhibition, arises from impaired axonal growth rather than defective innervation and arborization in the contralateral cortex.

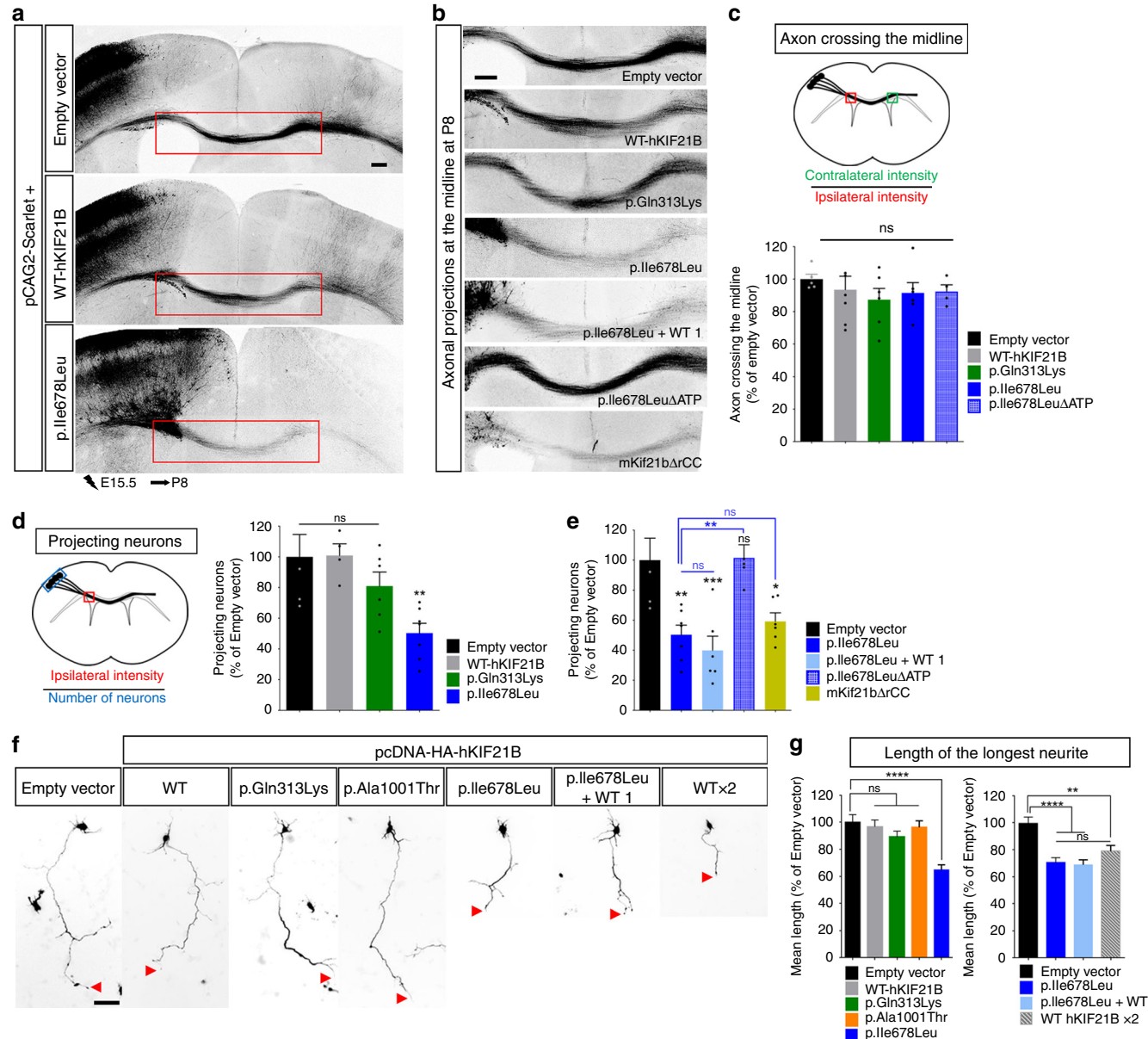

**Fig. 6 p.Ile678Leu hKIF21B variant impedes inter-hemispheric connectivity. a** Coronal sections of P8 brains after IUE with pCAG2-Scarlet and indicated NeuroD-IRES-GFP constructs. **b** Close-up views of the red boxed area in **a** showing impaired axonal inter-hemispheric connectivity upon expression of the p. Ile678Leu variant or the hyperactive mKif21bΔrCC but not upon expression of WT, p.Gln313Lys or immotile p.Ile678LeuΔATP. Rescue experiments were done by co-expressing p.Ile678Leu-hKIF21B (1 μg/μL) together with WT-hKIF21B constructs at 1 μg/μL (p.Ile678Leu + WT 1). Scale bars, 250 μm. **c**, **d** Upper (**c**) or left (**d**) panels, schematic describing methods used to quantify the percentage (**c**) of axon crossing the midline and (**d**) of projecting neurons. Lower (**c**) or right (**d**) panels, histograms presenting the percentage (**c**) of axon crossing the midline and (**d**, **e**) of projecting neurons. Data (means ± S.E.M.) were analyzed by one-way ANOVA (Bonferroni's multiple comparisons test). Number of pups analyzed: **c** Empty vector, $n = 5$; WT, $n = 7$; p.Gln313Lys, $n = 6$; p.Ile678Leu, $n = 6$; p.Ile678LeuΔATP, $n = 4$; **d**, **e** empty vector, $n = 5$; WT, $n = 4$; p.Gln313Lys, $n = 6$; p.Ile678Leu, $n = 7$; p.Ile678LeuΔATP, $n = 5$; p. Ile678Leu + WT, $n = 6$; mKif21bΔrCC, $n = 6$. **f** Representative DIV2 cortical neurons transfected at DIV0 with pCAG2-Scarlet together with empty pcDNA-HA or WT (at 1 (WT) or 2 μg/μL (WT ×2)) or mutant pcDNA-HA-hKIF21B constructs. Rescue experiments were done by co-expressing mutated p. Ile678Leu hKIF21B variant together with NeuroD-WT-hKIF21B (ratio 1:1; p.Ile678Leu + WT 1). Red arrowheads point to the axon tip. **g** Quantification of the longest neurite length (axon) at DIV2. Bars represent the means of the longest neurite length ± S.E.M. Significance was calculated by one-way ANOVA (Bonferroni's multiple comparisons test). Number of neurons analyzed: (left graph) Empty vector, $n = 94$; WT, $n = 96$; p.Gln313Lys, $n = 104$; p.Ile678Leu, $n = 144$; p.Ala1001Thr, $n = 133$, from four (Empty vector, WT, p.Ile678Leu) or three (p.Gln313Lys, p.Ala1001Thr) independent experiments; (right graph) empty vector, $n = 132$; WT 2 μg/μL, $n = 125$; p.Ile678Leu + WT, $n = 146$; p.Ile678Leu, $n = 134$; from four independent experiments. ns non-significant; *$P < 0.05$; **$P < 0.005$; ***$P < 0.001$; ****$P < 0.0001$. Scale bars, (**a**) 200 μm, (**b**) 250 μm (**f**) 50 μm. Source data are provided in the Source Data file.

**hKIF21B p.Ile678Leu variant impairs ipsilateral connectivity.** Callosal neurons not only branch contralaterally, but also send multiple ipsilateral axon collaterals within layer II-III and even more strongly to layer V. We further examined whether this

hKIF21B missense variant also impacts the establishment of intracortical connections. We performed IUE of WT and mutant hKIF21B in wild-type embryos at E15.5 and analyzed ipsilateral cortical collaterals at P8. Although callosal neurons expressing

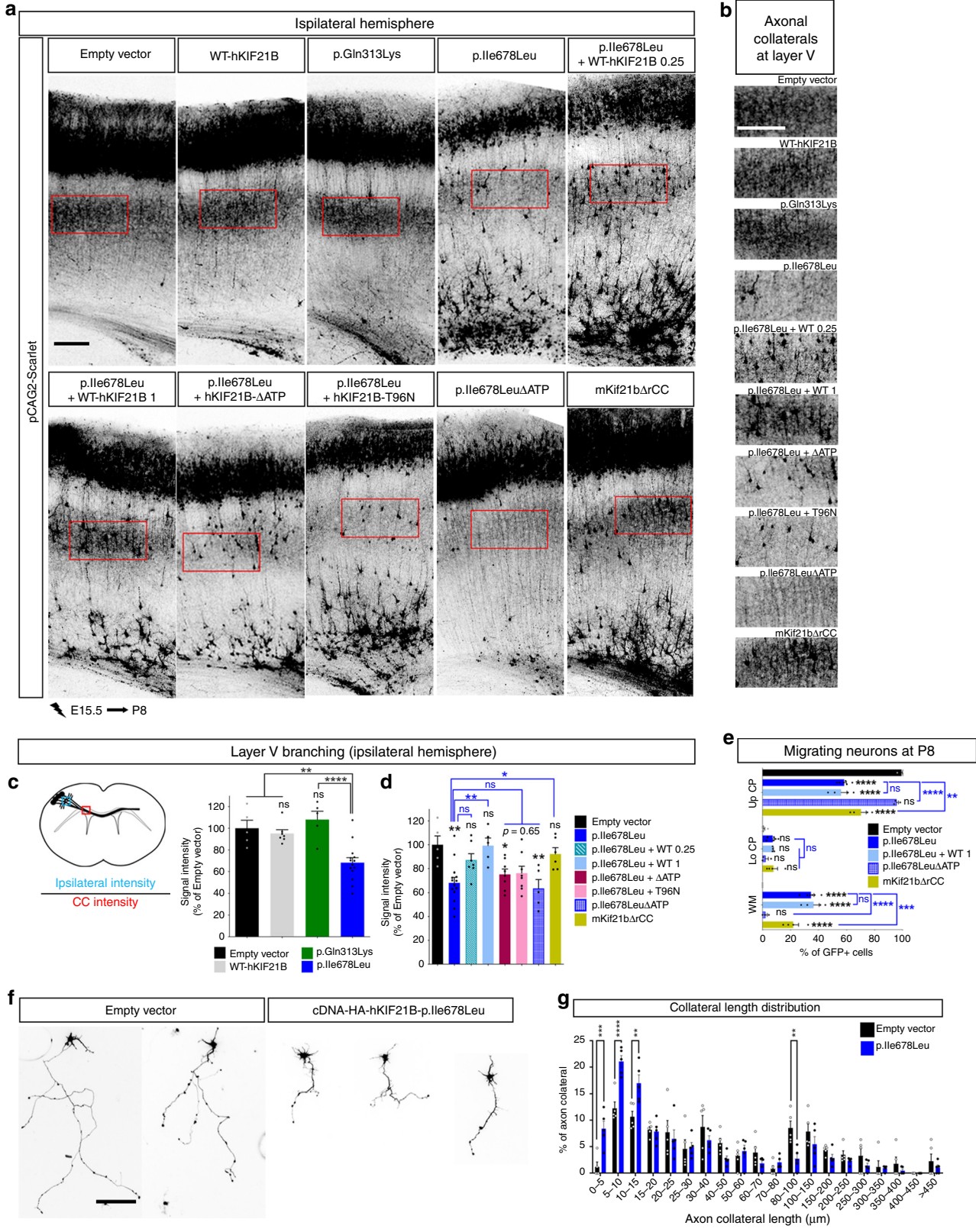

WT or p.Gln313Lys hKIF21B displayed prominent ipsilateral branching, overexpression of the p.Ile678Leu variant greatly reduced the intracortical branching (−32%, Bonferroni adjusted $P = 0.0016$) (Fig. 7a–c). Co-electroporation of p.Ile678Leu hKIF21B variant with either half or equivalent dose of WT-hKIF21B gradually restored the intrahemispheric connectivity

phenotype, suggesting that p.Ile678Leu hKIF21B possibly impairs formation of ipsilateral collaterals through a dominant negative mechanism (Fig. 7a, b, d). Notably, migration defects were not rescued in these experiments (Fig. 7e) demonstrating that the branching phenotype is not an indirect consequence of neuron mispositioning. Conversely, p.Ile678Leu-ΔATPhKIF21B

**Fig. 7 p.Ile678Leu hKIF21B variant impairs ipsilateral axon collaterals formation. a** Representative images of ipsilateral P8 cortices after IUE, with pCAG2-Scarlet and indicated NeuroD-IRES-GFP constructs. Rescue experiments were done by co-expressing mutated p.Ile678Leu hKIF21B together with increasing amount of WT-hKIF21B (at 0.25 μg/μL (WT 0.25) or 1 μg/μL (WT 1)) or equivalent amount of hKIF21B-ΔATP or hKIF21B-T96N, that both lost their processivity. **b** Close-up views of red boxed area in (**a**) showing axon ipsilateral branching within layer V for all the indicated conditions. **c**–**e** Histograms (means ± S.E.M.) showing (**c**, **d**) the quantification of the ipsilateral branching in layer V (intensity of scarlet signal in layer V (blue box) normalized on the intensity of the scarlet signal in the corpus callosum (red box) — as shown on the schematic in the upper panel) and (**e**) the distribution of electroporated neurons in three different regions (Up CP (upper cortical plate), Lo CP (lower cortical plate), and WM (White matter)). Data from were analyzed by (**c**, **d**) one-way ANOVA or (**e**) two-ways ANOVA (Bonferroni's multiple comparisons test). Number of pups analyzed: **c**, **d** Empty vector, $n = 5$; WT, $n = 7$; p.Gln313Lys, $n = 5$; p.Ile678Leu, $n = 14$; p.Ile678Leu + WT 0.25 μg/μL, $n = 7$; p.Ile678Leu + WT 1 μg/μL, $n = 5$; p.Ile678Leu + ΔATP, $n = 8$; p.Ile678Leu + T96N, $n = 8$; p.Ile678LeuΔATP, $n = 5$; mKif21bΔrCC, $n = 6$; **e** Empty vector, $n = 5$; p.Ile678Leu, $n = 6$; p.Ile678Leu + WT 1 μg/μL, $n = 4$; p.Ile678LeuΔATP, $n = 3$; mKif21bΔrCC, $n = 4$. **f** Representative DIV5 cortical neurons transfected at DIV2 with pCAG2-Scarlet together with the indicated pcDNA-HA-hKIF21B constructs. **g** Distribution (means ± S.E.M.) of axonal collateral branches length at DIV5. Significance was calculated by two-way ANOVA (Bonferroni's multiple comparisons test). Number of collaterals analyzed: Empty vector, $n = 265$; p.Ile678Leu, $n = 385$; from 51 (Empty vector) and 66 (p.Ile678Leu) neurons from five independent experiments. ns non-significant, $*P < 0.05$; $**P < 0.005$; $***P < 0.001$; $****P < 0.0001$. Scale bars, (**a**, **b**) 250 μm, (**f**) 150 μm. Source data are provided in the Source Data file.

overexpressing neurons that migrated normally (Fig. 7e), failed to send axonal collaterals ipsilaterally in layer V (Fig. 7a, b, d). Also, those neurons (p.Ile678Leu-ΔATPhKIF21B) showed normal axonal growth (Fig. 6b, c, e), ruling out the possibility that defective intracortical branching arises from impaired axonogenesis (Fig. 7a, b, d). Accordingly, branching defects likely arose from impaired collateral growth as revealed by in vitro analysis of axon branching in primary neurons at DIV5. Expression of p.Ile678Leu hKIF21B variant but not WT nor p.Gln313Lys or p.Ala1001Thr mutants led to a shift of branch length toward short branch classes that resulted in a large decrease of the mean length of axon collaterals (Fig. 7f, g, Supplementary Fig. 7a, c, d). Consistent with a dominant negative effect, intrahemispheric connectivity is not affected upon KIF21B hyperactivation with neurons expressing NeuroD-mKif21bΔrCC displaying normal ipsilateral collaterals (Fig. 7a, b, d). We finally investigated which function of WT-hKIF21B was negatively modulated by the mutant protein. We induced expression of p.Ile678Leu variant together with hKIF21B that either cannot bind (NeuroD-ΔATPhKIF21B; Fig. 1h) or hydrolyze ATP (T96N-hKIF21B; Fig. 1h)[21] at a 1:1 ratio using IUE at E15.5. Both constructs failed to rescue the branching phenotype induced by p.Ile678Leu hKIF21B at P8, suggesting that p.Ile678Leu hKIF21B exerts its dominant negative effect on the processive activity of KIF21B (Fig. 7a, b, d, Supplementary Fig. 3a). Collectively, our results showed that the p.Ile678Leu variant alters the intrahemispheric connectivity beyond its effect on migration and axonal growth through a dominant negative effect on motility.

## Discussion

Our findings highlight the critical role of KIF21B in the regulation of processes involved in cortical development and implicate variants in *KIF21B* in ID and brain malformation. We identified three missense variants and one duplication of four nucleotides. The duplication leads to a frameshift introducing a premature termination codon in exon 20. The resulting mutant mRNA is likely degraded by nonsense-mediated mRNA decay. Although we demonstrated in mice that *Kif21b* haploinsufficiency leads to an impaired neuronal positioning (Supplementary Fig. 3d–f), the p.Asn988Serfs*4 protein truncated variant is possibly not pathogenic. Indeed, h*KIF21B* gene might partially tolerate loss-of-function variants: in the gnomAD (Genome Aggregation Database, v2.1.1 "non-neuro") populations that is supposed to be depleted in severe pediatric conditions, 28 loss-of-function variants have been reported. Nonetheless, the ratio of the observed/expected loss-of-function variants in the gnomAD populations is

low (0.32, confidence interval 0.23–0.43), still questioning the penetrance of loss-of-function variants in *KIF21B*.

Our study provides the molecular mechanisms by which the identified variants lead to an abnormal brain phenotype. We showed that all missense variants, to various extents, impaired neuronal migration by enhancing KIF21B motor activity. Several lines of evidence suggest that hKIF21B missense variants exert a gain-of-function effect by enhancing KIF21B motility activity through lessening of the kinesin autoinhibition (Fig. 4). First, WT-hKIF21B is unable to rescue the variant-induced migratory defects at equivalent dose (Fig. 3a, b). Second, the phenotype induced by overexpression of the variants is phenocopied by the expression of a constitutively active form of KIF21B (that is truncated for the rCC domain) (Fig. 4c, d). Third, loss of ATP binding is sufficient to abrogate the phenotype induced by the missense variants (ΔATP-hKIF21B; Figs. 4i, j and 7e). Fourth, mutant hKIF21B proteins showed enhanced microtubule-based motility compared to the WT protein (Fig. 4g, h). How do hKIF21B variants lead to autoinhibition release? KIF21B autoinhibition is mediated by a regulatory segment (rCC) within the second coiled-coil domain (CC2, Fig. 1h) that fastens the CC2 domain to the motor head[42,46]. We hypothesize that the position of the missense variants within the motor (p.Gln313Lys), coiled-coil (p.Ile678Leu) and rCC (p.Ala1001Thr) domains (Fig. 1h) alters the protein conformation so that it varies the impact on the intramolecular interaction between the motor and the internal coiled-coil domains. This model raises the possibility that the level of disruption of these interactions correlates with the degree of autoinhibition release imposed by the different missense variants and therefore dictates the severity of the phenotype. In accordance, the increase in KIF21B processivity correlated with the extent of migration defects, the velocity of the p.Ile678Leu variant being the most drastically enhanced (Fig. 4g, h). We therefore propose a model in which a minimal level of autoinhibition is required to ensure proper function of KIF21B in the developing cortex. Below this threshold, the more KIF21B gets overactivated, the more severe and broad the phenotypes will present. In this model, the p.Gln313Lys and p.Ala1001Thr variants would partially relieve autoinhibition, whereas the p.Ile678Leu variant would completely loose autoinhibition. Consistently, p.Gln313Lys and p.Ala1001Thr KIF21B induce a delay of migration, whereas the p.Ile678Leu variant leads to a permanent arrest of migration and an additional connectivity phenotype (Figs. 3 and 6).

Beyond autoinhibition, maintaining a proper level of KIF21B activity seems to be crucial for its function during development. Indeed, *KIF21B* haploinsufficiency also leads to migratory defects. Whether those defects are caused by a loss of trafficking or MT

regulator functions is not clear and should be further assessed. Overall, the model could be expanded to a threshold of activity, below (haploinsufficiency) or above (identified missense variants) which KIF21B would not be properly functional leading to neurodevelopmental defects.

Convergent evidence suggests that the p.Ile678Leu variant alters axon branching through a dominant negative effect on KIF21B processivity (Fig. 7). First, co-expression of increasing amount of WT-hKIF21B together with the mutant protein gradually restored the intrahemispheric connectivity phenotype. Second, an immotile form of KIF21B failed to rescue the variant-induced branching defect. Third, expression of a constitutive active form (ΔrCC) of KIF21B does not affect the formation of axon branches. Fourth, p.Ile678Leu-ΔATPhKIF21B over-expressing neurons display abnormal ipsilateral collaterals. Fifth, consistent with a dominant negative effect, the WT KIF21B protein fails to form homodimer and is mislocalized when co-expressed with the p.Ile678Leu variant. Intriguingly, the p.Ile678Leu variant also perturbs axonal growth (Fig. 6) and migration (Figs. 3 and 4) through attenuation of autoinhibition, suggesting, as discussed above, a possible gain-of-function effect. To reconcile these seemingly conflicting findings, we propose that it could imply that KIF21B regulates the trafficking of different cargoes in axon and branches. p.Ile678Leu-induced over-activation of KIF21B might therefore lead to excessive motility of specific cargoes within axons. Conversely, the same variant could impede the transport of branch-specific cargoes by interfering with the function of the wild-type protein. At this time, none of the few KIF21B cargoes identified is specific to axon or branches[44,45], so further work is needed to identify cargoes in the different cellular compartments and validate this hypothesis.

Expression of the KIF21B p.Gln313Lys variant recapitulates the microcephaly phenotype observed in the reported subject. Given that introduction of the p.Ala1001Thr variant that is expected to attenuate KIF21B autoinhibition at the same extent as the p.Gln313Lys substitution variant, but does not reduce brain size in zebrafish (Fig. 5), the hyperactivation of KIF21B is unlikely to be driving the phenotype of microcephaly. In a search for possible microcephaly-underlying mechanism, we exclude any non-cell autonomous effect on progenitors' proliferation, or any impact on cell survival. Abnormal postnatal neuronal maturation may also contribute to the global microcephaly phenotype. Accordingly microcephaly may worsen with time: the patient carrying the KIF21B p.Gly313Lys variant was born with a head circumference (HC) of 32 cm (5th percentile), but microcephaly progressed and at the age of 12 was 48.5 cm (<1st percentile, −3.9 SD). These maturation defects may result from regulation of neuronal soma size[56,57] or from connectivity defects. Although the axonal branching is not impaired in callosal neurons overexpressing the microcephaly-related variant, we cannot rule out the possibility that reduced dendritic arborization of projection neurons influences the microcephaly phenotype.

In conclusion, the mechanism proposed here for the role of KIF21B in attenuating (p.Gln313Lys and p.Ala1001Thr) or abrogating autoinhibition (p.Ile678Leu) of one or several kinesin functions, might be expanded to other kinesins (KIF1A[28], KIF5C[30], KIF7[26], KIF4A[24]) known to be regulated by auto-inhibition and for which the pathophysiological mechanisms underlying the migration and inter-hemispheric connectivity phenotypes have not yet been elucidated. In addition, auto-inhibition of kinesins has been implicated in several physiological processes including the regulation of innervation, synaptogenesis and compartment-specific localization of cargo[53,54,58,59]. Our results indicate that fine-tuning of KIF21B activities is critical for proper neuronal migration and axonal growth, adding novel physiological roles of kinesin autoinhibition.

## Methods

**Whole-exome sequencing (WES).** A parent–offspring trio approach was used for whole-exome sequencing (WES) in each family. Exomes were sequenced using DNA isolated from blood according to standard procedures. Informed consent was obtained from all participants in accordance with site-specific institutional review board. Patient 1: The SeqCap EZ MedExome Enrichment Kit (Roche) was used for library preparation with 12 samples multiplexing, according to manufacturer's protocol. This library was then sequenced on a NextSeq 500 (Illumina) with a 2 × 150 bp high output flowcell. The bioinformatic analyses was conducted by Polyweb using BWA 0.7.12, picard-tools-1.121, GenomeAnalysisTK-2014.3- 17-g0583013, SNPEff-4.2. Patient 2: The SeqCap EZ VCRome 2.0 (Roche) was used for library preparation. Exome libraries were sequenced on an Illumina HiSeq 2500 instrument and the following sites are used to search for previously described gene pathogenic variants and polymorphisms: the Human Gene Mutation Database (HGMD), the single Nucleotide Polymorphism database (dbSNP), 1000 genomes, HapMap data. Patient 3: Exome capture was done using the Nimblegen SeqCap_EZ_Exome_v3 (Nimblegen). Exome libraries were sequenced on an Illumina HiSeq instrument (Illumina, San Diego, USA) with 150 bp paired-end reads at a median coverage of 100×. Sequence reads were aligned to the hg19 reference genome using BWA. Variants were subsequently called by the GATK unified genotyper, and annotated using a custom diagnostic annotation pipeline. Patient 4: Exome was captured using the Clinical Research Exome kit (Agilent Technologies, Santa Clara, CA). Massively parallel (NextGen) sequencing was done on an Illumina system with 100 bp or greater paired-end reads. Reads were aligned to human genome build GRCh37/UCSC hg19, and analyzed for sequence variants using GeneDx's XomeAnalyzer (a custom-developed variant annotation, filtering and viewing interface for WES data)[60]. The general assertion criteria for variant classification are publicly available on the GeneDx ClinVar submission page (http://www.ncbi.nlm.nih.gov/clinvar/submitters/26957/).

Inclusion and genetic studies were approved by local ethics committee in France (CCP Ile de France, CPP No. 71-10/ ID RCB: 2010-A00802-37) and USA (Institutional Review Board at Baylor College of Medicine, protocol H-29697 and at the John Hopkins School of Medicine).

**Cloning and plasmid constructs.** Wild-type (WT) human KIF21B cDNA (NCBI Reference Sequence: NM_001252100.1) was obtained from Vector Builder by gene synthesis and subcloned by restriction-ligation into the NeuroD-iresGFP[61], the pcDNA3.1+/N-HA and in the pcEGFP-N1 vectors. Myc-tagged hKIF21B constructs were obtained by replacing the GFP sequence of the pcEGFP-N1 containing the hKIF21B gene by a Myc-tag sequence. Human KIF21B variants c.937C>A (p.Gln313Lys), c.2032A>C (p.Ile678Leu), c.3001G>A (p.Ala1001Thr) and the c.288C>T (T96N) substitution that abolishes KIF21B mobility[21] were created from WT CDS by Sequence and Ligation Independent Cloning (SLIC). hKIF21B ATP binding site (amino acids 87–94; UniProtKB O75037) was deleted by SLIC from NeuroD-WT-hKIF21B-iresGFP, NeuroD-hKIF21B-p.Gln313Lys-iresGFP, NeuroD-hKIF21B-p.Ile678Leu-iresGFP and NeuroD-hKIF21B-p.Ala1001Thr-iresGFP to generate p.Ile678Leu-ΔATPhKIF21B WT-ΔATPhKIF21B, p.Gln313Lys-ΔATPhKIF21B, p.Ile678Leu-ΔATPhKIF21B and p.Ala1001Thr-ΔATPhKIF21B constructs, respectively.

Wild-type mouse Kif21b CDS was isolated from E18.5 cDNA mouse cortices by PCR and a new isoform has been amplified. This isoform is 4920 bp-long and has an insertion (c.4905_4906 ins C; NM_001039472.2) that leads to a frameshift and to the introduction of a premature stop codon in exon 35. This isoform has been subcloned by restriction-ligation into the NeuroD-IRES-GFP plasmid. This new isoform has also been fused to eGFP in the N-terminal part (pEGFP-C1-WT-mKif21b) via subcloning in pEGFP-C1 plasmid (NovoPro V12024). The amino acids 930-1010 corresponding to autoinhibitory domain (rCC)[42] were deleted by site-directed mutagenesis to generate NeuroD-mKif21b-ΔrCC construct. Mouse Kif21b 3′UTR sequence (NCBI Reference Sequence: NM_001039472.2) were amplified by PCR and cloned by restriction-ligation into the pEGFP-C1 plasmid and fused to eGFP in the N-terminal part. pCR-BluntII-TOPO-mKif21b 3′UTR used to synthesize RNA probes was generated by cloning part of the mKif21b 3′UTR (255 bp, Genepaint template T36548) in pCR-BluntII-TOPO vector.

shRNAs against coding sequence 3390–3410 (NM_001252100.1) (sh-Kif21b #1) or the 3′-UTR (sh-Kif21b #2) of mouse Kif21b were generated by annealing of sense and antisense oligos, the resulting duplex were subcloned in pCALSL-mir30[52] backbone vector digested with XhoI and EcoRI. The following oligos were used: sh-Kif21b #1:

sense:5′TCGAGaaggtatattgctgttgacagtgagcgCCACGATGACTTCAAGTTCAAtagtgaagccacagatgtaTTGAACTTGAAGTCATCGTGGgcctactgcctcgG 3′; antisense: 5′AATTCcgaggcagtaggcaCCACGATGACTTCAAGTTCAAtacatctg tggcttcactaTTGAACTTGAAGTCATCGTGGcgctcactgtcaacagcaatataccttC 3′; sh-Kif21b #2: sense: 5′TCGAGaaggtatattgctgttgacagtgagcgGCCTTTAACAACCAGAGTATAtagtgaagccacagatgtaTATACTCTGGTTGTTAAAGGCtgcctactgcctcgG 3′; antisense: 5′AATTCcgaggcagtaggcaGCCTTTAACAACCAGAGTATAtacatctg tggcttcactaTATACTCTGGTTGTTAAAGGCcgctcactgtcaacagcaatataccttC 3′; Scrambled shRNA: Sense: 5′TCGAGaaggtatattgctgttgacagtgagcgGCGCGATAGCGCTAATAAATTTtagtgaagccacagatgtaAAATTATTAGCGCTATCGCGCtgcc tactgcctcgG 3′; Antisense: 5′AATTCcgaggcagtaggcaGCGCGATAGCGCT

AATAATTTtacatctgtggcttcactaAAATTATTAGCGCTATCGCGCCgctcactgtcaa-cagcaatataccttC 3′

pCAG2-mScarlet and pSCV2-CAG-mVENUS[61] expressing vectors were provided by J. Courchet (INMG, Lyon, France). Plasmid DNAs used in this study were prepared using the EndoFree plasmid purification kit (Macherey Nagel).

**Mice**. All animal studies were conducted in accordance with French regulations (EU Directive 86/609 – French Act Rural Code R 214-87 to 126) and all procedures were approved by the local ethics committee and the Research Ministry (APA-FIS#15691-201806271458609). Mice were bred at the IGBMC animal facility under controlled light/dark cycles, stable temperature (19 °C) and humidity (50%) condition and were provided with food and water ad libitum.

Timed-pregnant wild-type (WT) NMRI (Janvier-labs) and CD1 (Charles River Laboratories) females were used for in utero electroporation (IUE) of sh-*Kif21b* and NeuroD-hKIF21B constructs, respectively, at embryonic day 14.5 (E14.5). Hybrid F1 females were obtained by mating inbred 129/SvJ females (Janvier-labs) with C57Bl/6J males (Charles River Laboratories). F1 females were crossed with C57Bl/6J males (Charles River Laboratories) to obtain timed-pregnant females for IUE at E15.5.

Kif21b KO mice were generated using the International Mouse Phenotyping Consortium targeting mutation strategy[62] and obtained from UC Davis/ KOMP repository (Kif21b^{tm1a(KOMP)Wtsi}). Genotyping was done as follows: Genomic DNA was extracted from tail biopsies using PCR reagent (Viagen) supplemented with Proteinase K (1 mg/mL), heated at 55 °C for 5 h. Proteinase K was inactivated for 45 min at 85 °C, and cell debris was removed by centrifugation. Samples were processed for PCR using the following primers: KIF21B forward: 5′-GGGGTACTT TCCATTGACCCAG-3′, KIF21B reverse: 5′-GAAGGGACCAAACCTGGGC-3′ for KIF21B targeted exon amplification and Mq forward 5′-GCTATGACTGGG CACAACAGACAATC-3′ and Mq reverse 5′-CAAGGTGAGATGACAGGAG ATCCTG-3′ for Neomycin gene amplification. The presence of the wild-type and knockout alleles was indicated by 346 and 261 bp products, respectively, which were detected on a 2% agarose gel.

**In utero electroporation (IUE)**. Timed-pregnant mice were anesthetized with isoflurane (2 L per min of oxygen, 4% isoflurane in the induction phase and 2% isoflurane during surgery operation; Tem Sega). The uterine horns were exposed, and a lateral ventricle of each embryo was injected using pulled glass capillaries (Harvard apparatus, 1.0 OD*0.58 ID*100 mmL) with Fast Green (1 µg/µL; Sigma) combined with different amounts of DNA constructs using a micro injector (Eppendorf Femto Jet). We injected 1 µg/µL of WT or mutant NeuroD-hKIF21B-IRES-GFP constructs together with 0.5 µg/µL of empty NeuroD-IRES-GFP vector at E14.5. 1.5 µg/µL of NeuroD-IRES-GFP vector were used as control. We injected 1 µg/µL of NeuroD:Cre-GFP vector together with 3 µg/µL of either Cre inducible pCALSL-miR30-shRNA-*Kif21b* #1 or #2 or pCALSL-miR30-sh-scramble sequence and 1 µg/µL of NeuroD- IRES-GFP or NeuroD-mKif21b-IRES-GFP (rescue experiment). For axonal pathfinding experiments, we injected 1 µg/µL of NeuroD-IRES-GFP (empty or containing WT or mutated human KIF21B cDNA) together with 0.8 µg/µL of pCAG2-mScarlet at E15.5. For rescue experiments, we co-injected 1 µg/µL of NeuroD-Ires-GFP (WT or mutated human KIF21B cDNA) together with 0.25, 0.5 or 1 µg/µL of the indicated NeuroD-IRES-GFP constructs. Plasmids were further electroporated into the neuronal progenitors adjacent to the ventricle by discharging five electric pulses (40 V) for 50 ms at 950 ms intervals using electrodes (diameter 3 mm; Sonidel CUY650P3) and ECM-830 BTX square wave electroporator (VWR international). After electroporation, embryos were placed back in the abdominal cavity and the abdomen was sutured using surgical needle and thread. For E16.5 and E18.5 analysis, pregnant mice were killed by cervical dislocation 2 and 4 days after surgery. For postnatal analysis, electroporated pups were killed 2, 4, or 8 days after birth (P2, P4, P8) by head sectioning or 22 days after birth (P22) by terminal perfusion.

**Mouse brain fixation, cutting and immunolabeling**. E12.5 to P8 animals were killed by head sectioning and brains were fixed in 4% paraformaldehyde (PFA, Electron Microscopy Sciences) in Phosphate buffered saline (PBS, HyClone) 2 h at room temperature (RT) or overnight (O/N) at 4 °C. P22 animals were killed by terminal perfusion of PBS then 4% PFA followed by overnight post-fixation at 4 °C in 4% PFA. For Kif21b expression pattern (Fig. 2 and Supplementary Fig. 2), immunolabeling was performed on cryosections as follows: after fixation, brains were rinsed and equilibrated in 20% sucrose in PBS overnight at 4 °C, embedded in Tissue-Tek O.C.T. (Sakura), frozen on dry ice and coronal sections were cut at the cryostat (12 to 18 µm thickness, Leica CM3050S) and processed for In situ hybridization or immunolabeling. Sections were maintained at −80 °C. For IUE analyses (Figs. 3–7, Supplementary Figs. 3–7), immunolabeling was performed on vibratome sections as follows: after fixation, brains were rinsed and embedded in a solution of 4% low-melting agarose (Bio-Rad) and cut into coronal sections (60-µm-thick for E18.5 and P2 mice, 100-µm-thick for P4 to P22 mice) using a vibrating-blade microtome (Leica VT1000S, Leica Microsystems) and processed for immunolabeling. Sections were maintained in PBS-azide 0,05% for short-term storage and in an Antifreeze solution (30% Ethyleneglycol, 20% Glycerol, 30% DH₂O, 20% PO₄ Buffer) for long-term storage.

For cryosections only, an antigen retrieval was performed by boiling sections in sodium citrate buffer (0.01 M, pH 6) during 15 min. Cryo- and vibratome sections were permeabilized and blocked with 5% Normal Donkey Serum (NDS, Dominic Dutsher), 0.1% Triton X-100 in PBS. Slides were incubated with primary antibodies diluted in blocking solution overnight at 4 °C and secondary antibodies diluted in PBS-0.1% Triton one hour at room temperature, whereas cell nuclei were identified using DAPI (1 mg/mL Sigma). Slices were mounded in Aquapolymount mounting medium (Polysciences Inc). All primary and secondary antibodies used for immunolabeling are described in Supplementary Table 1.

**RNA in situ hybridization**. Mouse *Kif21b* sense and antisense probes were synthesized from pCR-BluntII-TOPO-mKif21b 3′UTR by either *Bam*HI digestion followed by synthesis withT7 RNA polymerase (Roche) (sense probe) or *Eco*RV digestion followed by synthesis with SP6 RNA polymerase (Roche) (antisense probe). Kif21b digoxigenin-labeled probes using the DIG RNA labeling Kit SP6/T7 (Roche) according to the manufacturer's protocol. In situ hybridization was performed on E12.5 to E18.5 WT NMRI coronal embryonic brain cryosections as follow: sections were first rinsed in Phosphate buffered saline (PBS, HyClone) and dehydrated for 5 min in successive ethanol baths (70%, 95% and 100%) diluted in sterile milliQ water. The same amount of each antisense or sense Kif21b digoxigenin-labeled probes were diluted at 0.1 µg/µL in pre-warm Hybridization Buffer (4 M NaCl 0.2 M, 5 mM EDTA pH 8, 10 mM Tris-HCl pH 7.5, 10 mM NaH₂PO₄.2H₂O, 10 mM Na₂HPO₄, 2 mg/mL Ficoll, 2 mg/mL poly-vinylpyrrolidine, 2 mg/mL bovine serum albumin, 10% Dextran Sulfate, 1 mg/mL Yeast tRNA (ThermoFisher Scientific), 50% deionized formamide) and denaturated for 10 min at 70 °C. Sections were then incubated with the probe mix in a water-bath O/N at 70 °C in a sealed humidified chamber. Slides were then washed twice with the pre-warmed 1× Saline Sodium Citrate solution (SSC, 0.15 M NaCl, 15 mM Na Citrate) at 70 °C for 30 min and in the 0.2× SSC solution for one hour at 70 °C and for 5 min at RT. Slides were then washed twice in the MABT solution (0.1 M Maleic Acid, 0.15 M NaCl, 0.1% Tween-20, pH 6) for 30 min at RT under slow agitation and blocked in MABT solution supplemented with 2% blocking reagent (Roche) and 20% heat-inactivated Normal Donkey Serum (NDS, Dominic Dutsher) for 1 hour at RT. Slides were then incubated with the anti-DIG antibody (Abcam ab76907, 1:2500 diluted in blocking solution) O/N in humidified chamber at 4 °C. Slides were then washed four times in MABT for 15 min at RT then twice in Alkaline Phosphatase Buffer (NTMT, 100 mM NaCl, 100 mM Tris pH 9.5, 50 mM MgCl₂. H₂O, 0.1% Tween-20) for 10 min at RT and stained in NTMT supplemented with NBT (Sigma-Aldrich, 0.3 mg/mL) and BCIP (Sigma-Aldrich, 0.175 mg/mL) for 4 hours at RT in a box protected from the light. Reaction was stopped by transferring slides into PBS for 15 min, then slides were post-fixed in PFA4% diluted in PBS for 10 min at 4 °C. Slides were then washed 3 times in PBS for 10 min, air-dried and mount in Pertex mounting medium (Leica Micro-systems). Images were taken using a macroscope (Leica M420) connected to a Photometrics camera with the CoolSNAP software (v. 1.2).

**Primary neuronal culture, magnetofection and immunolabeling**. Cortices from E15.5 CD1 mouse embryos were dissected in cold PBS supplemented with BSA (3 mg/mL), MgSO4 (1 mM, Sigma), and D-glucose (30 mM, Sigma). Cortices were dissociated in Neurobasal media containing papain (20 U/mL, Worthington) and DNase I (100 µg/mL, Sigma) for 20 min at 37 °C, washed 5 min with Neurobasal media containing Ovomucoïde (15 mg/mL, Worthington), and manually triturated in OptiMeM supplemented with D-Glucose (20 mM). Cells were then plated at 2 × 10⁵ cells per 24-well plate coated with poly-D-lysine (1 mg/mL, Sigma) overnight at 4 °C and cultured for 2–5 days in Neurobasal medium supplemented with B27 (1×), L-glutamine (2 mM) and penicillin (5 units/mL)–streptomycin (50 mg/mL). To transfect cultured neurons, we performed magnetofection at DIV0 (for analysis at DIV2) or DIV2 (for analysis at DIV5) using NeuroMag (OZ Bioscience) according to the manufacturer's protocol. Cells were fixed at DIV2 or DIV5 for 15 min at room temperature in 4% PFA, 4% sucrose in PBS, and incubated for 1 h in 0.1% Triton X-100, 5% NDS in PBS. Primary antibodies were incubated over-night at 4 °C and secondary antibodies were incubated for 1 h at room temperature (see Supplementary Table 1 for antibodies). DNA was stained using DAPI (1/1000). Slides were air-dried and mounded in Aquapolymount mounting medium (Polysciences Inc).

**Fluorescent-activated cell sorting (FACS)**. Cortices from 3 to 4 E16.5 mouse Rosa26-loxSTOP-YFP; NEXCRE/+ embryos were dissected and dissociated as described above. After dissociation, cells were resuspended in 500 µL of staining solution (10% Fetal Bovine Serum (FBS) + 0.02% Sodium Azide in PBS) and stained for 20 min on ice in dark with CD24-APC Antibody (0.06 µg/100 µL final; clone M1/69, ThermoFisher Scientific (#17-0242-82)). Cells were then washed twice with HBSS (Gibco) and passed through a 40 µm filter (Filcon FACS). YFP-/CD24- population was sorted to enrich for progenitors and YFP⁺/CD24⁺ population was used to enrich for neurons using the BD Aria II flow cytometer with 488 and 633 lasers to excite YFP and APC respectively. Littermate YFP− embryos (Rosa26-loxSTOP-YFP; NEXCRE−) were processed the same way, stained with the Rat IgG2b-APC isotype control antibody (clone eB149/10H5, ThermoFisher Scientific (#17-4031-82)) and used as controls to set the YFP and CD24 gates.

**Microfluidic fabrication and neurons plating**. Design of polydimethylsiloxane microfluidic device is based on the one described by Taylor et al.[51] with modifications of the size of the microchannels (3-μm width, 3-μm height and 450-μm length to reduce the number of axons per microchannels)[63]. Briefly, microfluidics chambers were positioned and sealed on Iwaki boxes using plasma cleaner and then coated with poly-ᴅ-lysin (0.1 mg/mL) in the upper chamber, and with poly-ᴅ-lysin (0.1 mg/mL) and laminin (10 μg/mL) in the lower chamber. After overnight incubation at 4 °C microfluidic devices were washed two times with Neurobasal medium and once with growing medium (Neurobasal medium supplemented with 2% B27, 2 mM Glutamax, and 1% penicillin/streptomycin). Microchambers were then placed in the incubator until neurons were plated. Primary cortical neurons were prepared as follows: E15.5 C57Bl/6J mouse embryos were collected and cortices were dissected, followed by papain and cysteine digestion and trypsin inhibitor incubation. After mechanical dissociation, cortical neurons were resuspended in growing medium ($5 \times 10^6$ cells in 120 μL) and plated in the upper chamber with a final density of ~7000 cells/mm². Neurons were kept in the incubator for 1 h. Then, the two compartments were gently filled with growing medium.

Immunostaining in microchambers was performed from DIV5 culture[64]: after 30 min fixation in 4% PFA/Sucrose dissolved in PBS, all the compartments were blocked with a solution containing BSA 1%, normal goat serum 2%, Triton X-100 0.1%. For these two solutions, a bigger volume in the upper chamber was applied to create a pressure gradient. After 1 h incubation with blocking solution, neurites in both compartments were incubated overnight at 4 °C with primary antibody recognizing KIF21B and Tau. Secondary antibodies were added the following day for 4 h and microchambers were maintained in PBS for a few days in the dark at 4 °C (see Supplementary Table 1 for antibodies).

**RNA extraction, cDNA synthesis and RT-qPCR**. To assess Kif21b mRNA expression in mouse and zebrafish, total RNA was extracted from the cortices of WT NMRI mouse embryos or from whole zebrafish (*Danio rerio*) embryos (AB strain) at different time points of development, with TRIzol reagent (ThermoFisher Scientific). We used mKif21b ex2-3 and drKIF21B ex2-3 primers to target *mKif21b* or *zKif21b* cDNA and mGAPDH or drElfA (Elongation factor 1-alpha) as housekeeping genes normalizer (Table 2). shRNA-*Kif21b* knock-down efficacy was assessed by RT-qPCR. Total RNA was prepared from HEK293T cells overexpressing sh-scrambled or shRNA-*Kif21b* #2 together with pEGFP-C1-3′UTR mKIF21B or from HEK 293K cells overexpressing sh-scrambled or shRNA-*Kif21b* #1 together with pEGFP-C1-WT-mKif21b. We used GFP primers to target the 3′ UTR sequence of *mKif21b* cDNA fused to GFP and mKIF21B ex2-3 primers to target *mKif21b* cDNA. We used mGAPDH or Hprt1 primers as housekeeping genes normalizer (Table 2). cDNA samples were synthetized with SuperScript IV Reverse Transcriptase (Invitrogen) and submitted to DNAse I treatment (TurboDNAse, ThermoFisher). RT-qPCR was performed in a LightCycler PCR instrument (Roche) using SYBR Green Master Mix (Roche). RT-qPCR was also performed to assess the degradation of the mutant mRNA by nonsense-mediated decay (NMD). A blood sample was obtained from patient 4 in PAXgene Blood RNA Tube (Qiagen, Germantown, CA, USA) and total RNA extracted using Qiagen PAXgene Blood miRNA kit (Qiagen, Germantown, CA, USA). As controls, total RNA was prepared with TRIzol reagent (ThermoFisher Scientific) from blood sample obtained three male individuals aged between 3 and 8 years who do not carry any *KIF21B* variants in PAXgene Blood RNA Tube. We used hKIF21B ex2-3 and hKIF21B ex33-34 coupled primers to target *hKIF21B* cDNA and hTBP as housekeeping gene normalizer.

**Cell culture, transfections and immunolabeling**. All cells used in this study are provided by the cell culture platform of the IGBMC (Strasbourg), are guaranteed mycoplasma free (PCR test Venorgem) and have not been authenticated. Mouse neuroblastoma N2A (ATCC) cells and Cos7 (ATCC) cells were cultured in DMEM (1 g/L glucose, GIBCO) supplemented with 5% Fetal Calf Serum (FCS) and Gentamicin 40 μg/mL in a humidified atmosphere containing 5% CO₂ at 37 °C. Human embryonic kidney (HEK) 293T cells were cultured in DMEM (1 g/L glucose) (GIBCO) supplemented with 10% FCS, penicillin 100 UI/mL, streptomycin

100 μg/mL in a humidified atmosphere containing 5% CO₂ at 37 °C. Mouse ST cells are neuronal progenitor cell lines from E14 striatal primordia of WT embryos immortalized using tsA58 SV40 large T antigen[65]. ST cells were cultured DMEM (1 g/L glucose) supplemented with 10% FCS heat-inactivated, non-essential amino acids, penicillin 100 UI/mL, streptomycin 100 μg/mL and G418 400 μg/mL in a humidified atmosphere containing 5% CO₂ at 33 °C. Cells were transfected using Lipofectamine 2000 (Invitrogen) according to the manufacturer's protocol. Expression of transfected genes was analyzed 48 h after transfection by immunoblotting. For localization experiments, ST cells were fixed 48 h after transfection for 5 min in −20 °C MeOH/AcOH solution (1:1), and incubated for 1 h in 0.5% Triton X-100, 5% NDS in PBS. Primary antibodies were incubated overnight at 4 °C and secondary antibodies were incubated for 1 h at room temperature (see Supplementary Table 1 for antibodies). DNA was stained using DAPI (1/1000). Slides were air-dried and mounted in Aquapolymount mounting medium (Polysciences Inc).

**Immunoprecipitation**. Immunoprecipitation (IP) experiments were done using the Pierce Anti-Myc Magnetic Beads kit (ThermoScientific) according to the manufacturer's protocol. HEK293T cells were transfected with the indicated HA-tagged, GFP-tagged and Myc-tagged constructs (ratio 1:1:2) using X-treme GENE 9 DNA transfection reagent (Roche). Cells were lysed 24 h after transfection in ice-cold IP buffer from the kit, supplemented with EDTA-free protease inhibitors (cOmplete™, Roche) and 0.01 M phosphatase inhibitor PMSF, for 30 min on ice. Cells debris were removed by high speed centrifugation at 4 °C for 15 min. After protein concentration measurement, samples were diluted to 1 μg/μL. Half of the protein were kept (Input) and diluted in 2× Laemmli Elution Buffer (Bio-Rad) containing 2% β-mercaptoethanol. 400 μg of proteins were then incubated with 15 μL of prewashed magnetic Myc-coupled beads for 2 h at 4 °C under gentle shaking. Beads were collected using a magnetic stand and supernatants were discarded. Beads were then washed twice with IP buffer and proteins were eluted at 95 °C by adding 2× Laemmli Elution Buffer containing 2% β-mercaptoethanol. The same volume of sample for all condition were loaded and analyzed by western blot as described below.

**Protein extraction and western blot**. Proteins from mouse cortices (E14.5 to P2) or from transfected cells (N2A, HEK293T, Cos7 and ST cells) were extracted as follows: cells were lysed in RIPA buffer (50 mM Tris pH 8.0, 150 mM NaCl, 5 mM EDTA pH 8.0,1% Triton X-100, 0.5% sodium deoxycholate, 0.1% SDS) supplemented with EDTA-free protease inhibitors (cOmplete™, Roche) for 30 min, then cells debris were removed by high speed centrifugation at 4 °C for 25 min. Protein concentration was measured by spectrophotometry using Bio-Rad Bradford protein assay reagent. Samples were denatured at 95 °C for 10 min in Laemmli buffer (Bio-Rad) supplemented with 2% β-mercaptoethanol and then resolved by SDS–PAGE and transferred onto nitrocellulose membranes. Membranes were blocked in 5% milk in PBS buffer with 0.1% Tween (PBS-T) and incubated overnight at 4 °C with the appropriate primary antibody in blocking solution. Membranes were washed three times in PBS-T, incubated at room temperature for 1 h with HRP-coupled secondary antibodies (Invitrogen) at 1:10,000 dilution in PBS-T, followed by three times PBS-T washes. Visualization was performed by quantitative chemiluminescence using SuperSignal West Pico PLUS Chemiluminescent Substrate (Sigma). Signal intensity was quantified using ImageQuant LAS 600 (GE Healthcare). Primary and secondary coupled HRP antibodies used for western blot are described in Supplementary Table 1. Relative protein expression was quantified using ImageJ software (Java 1.8.0_112).

**Cycloheximide (CHX) treatment**. To assess the protein half-life of WT- and mutated hKIF21B proteins, treatments using the translational inhibitor cycloheximide (CHX) were performed. N2A cells were cultured on 6-well plates and transfected with the adequate NeuroD- hKIF21B constructs using Lipofectamine 2000 (Invitrogen) according to the manufacturer's protocol. The day after, cells were treated with CHX (Sigma) diluted in media at 10 μg/mL for either 2, 4, 6, 8 or

**Table 2 List of primers used for RT-qPCR.**

| Gene | Specie | Forward sequence | Reverse sequence |
|---|---|---|---|
| mKIF21B ex2-3 | Mouse | AAGGCTGCTTTGAGGGCTAT | AAAGCCGGTGCCCATAGTA |
| hKIF21B ex3-4 | Human | GTCACTTCTCGCCTCATCCA | CTCTGCACGTTCATCTGGGT |
| hKIF21B ex33-34 | Human | TCAGTGGCTCCCGAGATAAC | CTTGTGCGCATTGGGGATTT |
| drKIF21B ex2-3 | Zebrafish | TCATCGAGGGCTGCTTTGAG | GACACGCTCACGTCAAAACC |
| GFP | | TACGGCAAGCTGACCCTGAAGT | GAAGTCGTGCTGCTTCATGTGG |
| mGAPDH | Mouse | TGATGACATCAAGAAGGTGGTGAAG | TCCTTGGAGGCCATGTAGGCCAT |
| drElfA | Zebrafish | CTTCTCAGGCTGACTGTGC | CCGCTAGCATTACCCTCC |
| hTBP | Human | CGGCTGTTTAACTTCGCTTC | CACACGCCAAGAAACAGTGA |
| Hprt1 | Human | AGGCGAACCTCTCGGCTTTC | TCATCATCACTAATCACGACGCC |

10 h. Cells were lysed as described above. For analysis, 10 µg of protein of each sample were loaded on a SDS-gel followed by western blotting analysis as described above. Experiments consisted of at least three independent replicates. Relative protein expression was quantified using ImageJ software (Java 1.8.0_112).

**Zebrafish manipulation**. Zebrafish (*D. rerio*) embryos (AB strain) maintenance and experiments were performed as described here https://zfin.org/zf_info/zfbook/cont.html#cont1. Human WT and mutant full-length cDNA were cloned into pCS2 vector and transcribed using the SP6 Message Machine kit (Ambion). We injected 1 nL of diluted RNAs (WT or mutants) at 100 ng/µL into wild-type zebrafish eggs at 1- to 2-cell stage. Length of the head was measured at 5 dpf as shown by the double arrow headed red lines (Fig. 5a). All the experiments were repeated at least three times and an unpaired two-tailed Student's *t*-test or a Welch's two samples *t*-test (when the variances are unequal) was performed to determine significance. All images were taken using a macroscope (Leica M420) dedicated to brightfield acquisitions equipped with a Coolsnap CF camera controlled by Coolsnap software. To perform zebrafish whole-mount immunolabeling, larvae were fixed at 5 dpf in Dent's fixative (80% methanol, 20% dimethylsulfoxide [DMSO]) overnight at 4 °C. The larvae were rehydrated slowly by decreasing concentrations of methanol. Larvae were then washed in PBS buffer with 0.1% Tween (PBS-T). After bleaching for 30 min in (10% $H_2O_2$ in PBS-T with KOH (0.5 g/mL), the larvae were rinsed in PBS-T, twice for 10 min each. Then larvae were permeabilized with proteinase K, then post-fixed with 4% PFA and washed in PBS-T. PFA-fixed larvae were washed in IF buffer (0.1% Tween-20, 1% BSA in PBS) for 10 min, then incubated in the blocking buffer (10% FBS, 1% BSA in PBS) for 1 h at room temperature. Larvae were incubated with primary antibody diluted in blocking solution, overnight at 4 °C (see Supplementary Table 1 for antibodies). After two washes in IF Buffer for 10 min each, larvae were incubated in secondary antibody diluted in blocking solution 2 h at room temperature. Larvae were washed in IF buffer for 10 min, twice. All images were taken using an epifluorescence macroscope at ×5 magnification equipped with a Coolsnap CF camera controlled by Coolsnap software v1.2.

**Image acquisition and analysis**. Cell counting and in vivo branching analyses were done in at least three different brain slices of at least three different embryos or pups for each condition. After histological examination, only brains with comparative electroporated regions and efficiencies were conserved for quantification.

For neuronal migration analyses, proliferation analyses and brain immunofluorescence experiments, a z stack of 1.55 µm was acquired in 1024 × 1024 mode using a confocal microscope (Leica TCS SP5 equipped with an hybrid camera and a HC PL APO ×20/0.70 objective) controlled by Leica Las X software v3.7 and analyzed using ImageJ software (Java 1.8.0_112). Cortical wall areas (upper cortical plate (Up CP), lower cortical plate (Lo CP), intermediate zone (IZ), subventricular zone (SVZ)/ventricular zone (VZ)) were identified according to cell density (nuclei staining with DAPI). The total number of GFP-positive cells in the embryonic brain sections was quantified by counting positive cells within a box of fixed size and the percentage of positive cells in each cortical area was calculated. For proliferation analyses, the total number of markers-positive cells in embryonic brain sections was quantified by counting positive cells in the intermediate zone (IZ) and in the subventricular zone (SVZ) below the electroporated region using square of 100-µm width or 50-µm width, respectively, with anatomically matched positions in experimental groups.

For in vivo branching analyses, a z stack of 3 µm was acquired in 1024 × 1024 mode using confocal microscope (Leica TCS SP5 equipped with an hybrid camera and a PL FL ×10/0.30 objective) and analyzed using ImageJ software (Java 1.8.0_112). For axonal midline crossing analyses, the fluorescence within a box of fixed size placed at the contralateral side of the midline of the brain section was measured and divided by the fluorescence at the ipsilateral side of the midline area (Fig. 6c). This value was normalized to empty vector (control) value. For projecting neurons analyses, the fluorescence intensity within a box of fixed size placed at the ipsilateral side of the midline of the brain section was measured using ImageJ (Java 1.8.0_112) and this value was divided by the number of Scarlet-positive neurons within a box of fixed size placed at the upper cortical plate (Fig. 6d). For quantification of layer V (ipsilateral) collateral branching, a box of fixed size was drawn encompassing layer V and the fluorescence area within this box was measured. This value was then divided by the fluorescence at the ipsilateral side of the midline area (Fig. 7c). We quantified contralateral branching[66] as follows: fluorescence within a box of fixed size placed contralaterally is measured on a vertical axis (from ventricular boundaries to pial surface) and converted from pixel size to percentile (0%: ventricular zone, 100%: pial surface) using the ProfilePlot plugin from FIJI software. Signal intensity of 100% is set as the maximum signal intensity in migration percentile 0 to 20 (considered as the white matter).

For in vitro branching analyses, primary neuronal cultures were done independently at least two (DIV5) or three (DIV2) times. DIV2 and DIV5 axonal length measures were performed in 125–146 and 30–66 independent cells respectively. For DIV5 axon collateral length distribution analyses, measures were performed in 187–385 collaterals from 30 to 66 independent cells. Images were acquired using upright fluorescence (DM 4000B; Leica) equipped with a HC PLAN APO20/×0.70 objective and a CoolSnap FX monochrome camera (Photometrics)

controlled by Leica Las X software v3.7. To measure the axonal length (longest neurite) at DIV2 and DIV5, the longest Scarlet-positive-labeled neurite was traced and the length was measured using Simple Neurite Tracer plugin (https://imagej.net/Simple_Neurite_Tracer) from FIJI software. Branch number and length was measured at DIV5 using Simple Neurite Tracer plugin.

For variant subcellular localization analysis (Fig. 4e, f), a z stack of 0.13 µm was acquired in 1024 × 1024 mode using a confocal microscope (Leica TCS SP5 equipped with an hybrid camera, using a HCX PL APO ×63/1.40–0.60 oil objective) controlled by Leica Las X software v3.7. For each condition, 67 to 229 independent cells from three independent transfections were segregated according to the subcellular localization of HA-tagged constructs (diffuse localization versus impaired localization).

**Live-cell imaging procedure and analysis**. Cos7 cells were grown on 35-mm glass bottom microwell dishes No. 0 (MatTek, U.S.A). Cells were transfected with the different pcEGFP-N1-hKIF21B constructs or co-transfected with BDNF-mCherry (gift from Gary Banker, Oregon Health and Science University, Portland, USA) or Mito-RFP (gift from Hélène Puccio, IGBMC, Illkirch, France) and the different pcDNA3.1+/N-HA-hKIF21B constructs or empty vector with a DNA ratio of 1:2 as described above. Live-cell imaging was done 24 h after transfection. Live videomicroscopy was performed on an inverted microscope Leica CSU W1 DMI8 (Leica) with an Adaptative Focus Control (AFC) controlled by Metamorph software v 7.6, using an HCX PL APO Lambda blue ×63/1.40 oil objective or an HC PL APO 100 × 1.47 oil objective. The microscope and the chamber were kept at 37 °C and 5% $CO_2$. Images were collected using an Orca Flash 4.0 camera with an exposure time of 80–150 ms. Images were acquired every 250 ms for 30 s to 1 min. Single GFP-positive kinesin were tracked using Manual Tracking plugin (https://imagej.nih.gov/ij/plugins/track/track.html) from Fiji software. For each condition, 182–203 kinesin particles in 25–32 independent cells were analyzed from at least three independent transfections. BDNF vesicles velocities and RFP-positive mitochondria were obtained from kymograph analysis, using FIJI plugin Kymo Tool-Box v.1.01 (https://github.com/fabricecordelieres/IJ-Plugin_KymoToolBox/releases). For each condition, 250–322 BDNF particles in 10–19 independent cells were analyzed from at least three independent transfections. For each condition, 173–256 Mito-RFP particles in 19–31 independent cells were analyzed from four independent transfections.

**Statistics and reproducibility**. Immunofluorescence, in situ hybridization, FACS-sorting and western blot experiments (Figs. 2c–e and 3e; Supplementary Figs. 2a–d, 3c, 4c, 5b) were repeated three times independently and gave similar results. Experiments on Kif21b WT and knockout mouse brain (Supplementary Fig. 2e, f) were performed on $n = 3$ and $n = 5$ different brains per genotype, respectively, and gave similar results. Western blot expression profile of NeuroD-IRES-GFP constructs (Supplementary Fig. 3a), HA-tagged, Myc-tagged (Supplementary Fig. 4a) and GFP-tagged constructs (Supplementary Fig. 4b) were performed once. For in utero electroporation experiments, only brains with comparative electroporated regions and efficiencies were conserved for quantification and statistical analyzes. Analysis of IUE experiments were performed blinded. For one embryo or pup brain, cell counting and in vivo branching analyses were performed in three different slices. The exact numbers (*n*) of samples, animals, cells, particles, axons or branches used to derive statistics are mentioned in figure legends along with the respective data and are also reported in Supplementary Data 1. The number of times each experiment was repeated independently (i.e the number of independent transfection or magnetofection) and statistical tests are mentioned in figure legends along with the respective data whenever possible, and are also reported in Supplementary Data 1. Statistical details (adjustments made for multiple comparisons, confidence intervals and exact *P*-values) for Figs. 2a; 3b, d; 4b, d, e—right panel, g, h, j; 5b; 6d, e, g; 7c–e, g; Supplementary Figs. 3d, f; 4d—right panel, f, g; 6d; 7b–d, are reported in Supplementary Data 1. All statistics were calculated using Prism (GraphPad, version 6) and are represented as mean ± S.E.M. Graphs were generated using Prism and images were assembled with Adobe Photoshop 13.0.1 (Adobe Systems).

**Reporting summary**. Further information on research design is available in the Nature Research Reporting Summary linked to this article.

## Data availability

The source data underlying Figs. 2a, b; 3b, d; 4b, d, e, g, h, j; 5b; 6c, d, e, g; 7c, d, e, g; and Supplementary Figs. 1d; 2d, f; 3a, b, d, f, h; 4a, b, d, f, g, h, i; 5a, b, d, g, j; 6c, d, h; 7b, c, d are provided as a Source Data file. All other relevant data included in the article are available from the authors upon request. The following databases and in silico software were used in the study: Human Gene Mutation Databases (http://www.hgmd.cf.ac.uk/ac/introduction.php?lang=english), the single Nucleotide Polymorphism database (http://ftp.ncbi.nih.gov/snp/), genome aggregation database (gnomAD, https://gnomad.broadinstitute.org), 1000 genomes (https://www.internationalgenome.org/), Polyphen-2 (http://genetics.bwh.harvard.edu/pph2/), Mutation Taster (http://www.mutationtaster.org/), Sorting Intolerant from Tolerant (SIFT, https://sift.bii.a-star.edu.sg/) and Combined Annotation Dependent Depletion (CADD, https://cadd.gs.washington.edu/).

The three h*KIF21B* missense variants have been deposited in LOVD (Leiden Open Variation Database) v3.0 (https://databases.lovd.nl/shared/genes/KIF21B) under the accession numbers 0000663938 (p.Ile678Leu), 0000663939 (p.Gln313Lys) and 0000663940 (p.Ala1001Thr).

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

## Acknowledgements

This work was funded by grants from INSERM (ATIP-Avenir program, J.D.G.), the Fyssen foundation (J.D.G.), the French state funds through the Agence Nationale de la Recherche under the project JCJC CREDO ANR-14-CE13-0008-01 (J.D.G.), CILAXCAL (C.D.) and AXYON ANR-18-CE16-0009-01 (F.S.), and the program Investissements d'Avenir labeled (ANR-10-IDEX-0002-02, ANR-10-LABX-0030-INRT, to J.D.G. and C.G.), the Fondation pour la recherche sur le cerveau (F.S.), INSERM/CNRS and University of Strasbourg. N.A.H. and G.Z. are supported by USDA Grant Number 3092-51000-057-04S. L.A. and J.R.A. are funded through the IGBMC PhD program (ANR-10-IDEX-0002-02, ANR-10-LABX-0030-INRT). L.A is currently supported by Fondation pour la recherche médicale (FDT201805005184). P.T. and C.W. are, respectively, research assistant and research engineer at the University of Strasbourg. J.D.G. and C.G. are INSERM investigators. F.S. is a professor at Univ. Grenoble Alpes. H.V. is supported by a PhD fellowship from Association Huntington France. We thank the Imaging Center of IGBMC (ici.igbmc.fr), in particular, Elvire Guiot and Erwan Grandgirard for their assistance in the imaging experiments. We are grateful to the staff of the mouse facilities of the Institut Clinique de la souris (ICS) and Institut de Génétique et de Biologie Moléculaire et Cellulaire (IGBMC), the staff of the zebrafish facility of the IGBMC and, the molecular biology service (in particular Thierry Lerouge and Paola Rossolillo) for their involvement in the project. We thank Sandra Bour and IGBMC communication service. We also thank Dr Courchet, Dr Banker, Dr Puccio and member of Chelly lab for sharing reagents and for discussion. We are really grateful to Pr Jamel Chelly and Dr Laurent Nguyen for their continuous support, discussion and time reading this manuscript. We warmly thank Dr Sandrine Humbert and Dr Binnaz Yalcin for helpful comments and advices. We are also grateful to members of J.D.G. and Chelly laboratories for discussion and technical assistance. In particular, we thank Dr Efil Bayam for cell sorting and advices in writing the manuscript. We thank Dr Gabrielle Rudolf for reading the manuscript and for her suggestions. We finally thank Paula Hernandez for her help collecting patient samples.

## Author contributions

L.A. and J.R.A. conceived and designed the experiments, performed the experiments, performed statistical analysis and analyzed the data related to cellular, and functional studies in mice. P.T. provided technical assistance and performed in utero electroporation. C.G. and C.S.B. conceived, designed and performed experiments in zebrafish. C.W. provided technical assistance for zebrafish studies. S.H., K.B., C.A.B., A.C., A.D., S.M., L.F., N.A.H., K.M., C.M., H.S., C.T.R., M.M.W., P.J.G.Z., G.Z. and D.H. contributed clinical and imaging data and follow-up of patients and families. S.H., A.R., C.N. and C.D. contributed to the generation of whole-exome sequencing, bioinformatics tools and analysis of sequencing data. H.V. and F.S. conceived and performed the expression analysis in microdevices. J.D.G. conceived, coordinated and supervised the study, designed experiments, analyzed data and wrote the manuscript.

## Competing interests

A.D. and K.M. are employees of GeneDx, Inc. The other authors declare no competing interest.

## Additional information

[1]Institut de Génétique et de Biologie Moléculaire et Cellulaire, Illkirch, France. [2]Centre National de la Recherche Scientifique, UMR7104, Illkirch, France. [3]Institut National de la Santé et de la Recherche Médicale, INSERM, U1258 Illkirch, France. [4]Université de Strasbourg, Strasbourg, France. [5]Département de Génétique, AP-HP, Hôpital de la Pitié-Salpêtrière, Paris, France. [6]Groupe de Recherche Clinique (GRC) "Déficience Intellectuelle et Autisme", UPMC, Paris, France. [7]Centre de Référence Déficiences Intellectuelles de Causes Rares, Hôpital de la Pitié-Salpêtrière, Paris, France. [8]Univ. Grenoble Alpes, INSERM, U1216, CHU Grenoble Alpes, Grenoble Institut Neuroscience, Grenoble, France. [9]Department of Molecular and Human Genetics, Baylor College of Medicine, Houston, TX, USA. [10]Texas Children's Hospital, Houston, TX, USA. [11]Department of Neurogenetics, Kennedy Krieger Institute, Baltimore, MD 21205, USA. [12]GeneDx, Gaithersburg, MD 20877, USA. [13]Centre de Référence Anomalies du Développement et Syndromes Malformatifs, Fédération Hospitalo-Universitaire Médecine Translationnelle et Anomalies du Développement (TRANSLAD), Centre Hospitalier Universitaire Dijon et Université de Bourgogne, Dijon, France. [14]Equipe GAD, INSERM LNC UMR 1231, Faculté de Médecine, Université de Bourgogne Franche-Comté, Dijon, France. [15]Department of Neurology, Boston Children's Hospital, Boston, MA 02115, USA. [16]INSERM, U 1127, CNRS UMR 7225, Faculté de Médecine de Sorbonne Université, UMR S 1127, Institut du Cerveau et de la Moelle épinière, ICM, Paris, France. [17]Centre de Référence Déficiences Intellectuelles de Causes Rares, Centre Hospitalier Universitaire Dijon, Dijon, France. [18]Department of Clinical Genetics, Amsterdam UMC, Vrije Universiteit Amsterdam, Amsterdam, The Netherlands. [19]Institute of Human Genetics, University Hospital Essen, University of Duisburg-Essen, Essen, Germany. [20]These authors contributed equally: José Rivera Alvarez, Solveig Heide, Camille S. Bonnet. ✉email: godin@igbmc.fr

