## [Peer Review File · Nature Communications]

Reviewers' comments:

Reviewer #1 (Remarks to the Author):

In this paper the authors have identified several KIF21B amino-acid change mutations from human intellectual/neurodevelopmental disability patients with partial changes in MRI, and then characterized their way of dysfunction using in vitro and in vivo model systems of neuronal migration and axonal outgrowth. Both of the motor domain mutation and coiled-coil domain mutation gave out significant phenotypes, but only the former one could be rescued by wild-type cDNA overexpression. The reason of this discrepancy was speculated that the latter one would impair the autoinhibition so that it might be a gain-of-function mutant. This manuscript will merit for publication because it may provide one of the first experimental evidence that KIF21B mutation is responsible for human neurodevelopmental abnormality in the CNS. The weak points are [1] that the cell biological investigation on the causal relationship (such as the cargo identification and motor activity assay) is insufficient and [2] that the intellectual/behavioral abnormality was not assessed using the genetic models. Accordingly, this reviewer will request the authors to improve it along the direction of [1] and [2], to meet the quality of this class of journals.

Specific points:

1. The authors should assess the possible changes in microtubule-based motor ATPase activity of wild type and mutant KIF21B proteins to see whether the autoinhibition is really affected. The easiest assay to accomplish this will be transfecting each GFP-tagged construct to fibroblasts to see the difference in distribution/motility (Zhao et al., Cell 2001).
2. What is the most responsible binding partners of KIF21B for these phenotypes? The speculations in the Discussion section should be actually tested by cell biological analyses.
3. Do these mutant constructs also affect the cognitive flexibility in behavioral tests? It might be best to generate the mutant model mice with CRISPR/Cas9 technology, but a less time-consuming way of testing this will be injecting AAV-mediated expression vectors directly into hippocampus to see whether contextual fear extinction is affected (Morikawa et al., Cell Rep 2018).
4. Fig. 1e–g. The methods of weighted imaging in the MRI study appeared different from patient to patient. This should be unified for comparison.

Reviewer #2 (Remarks to the Author):

In this manuscript Asselin et al provide initial evidence that likely pathogenic variants in the KIF21B gene could cause a cognitive phenotype and brain abnormalities.

Significant Criticisms:

1. The evidence that the frameshift denovo variant in patient 4 is causative is not at all convincing. KIF21B has a PLI of 0 and therefore haploinsufficiency is likely to be tolerated. There are numerous truncating alleles in gnomAD and BRAVO apparently not related to a cognitive phenotype. In addition, the Kif21B1 +/- KO heterozygotes mice do not show a phenotype. This reviewer strongly suggests to delete patient 4 and the part of the paper related to this case.
2. Since the PLI of KIF21B is 0, the expectation is that the non-synonymous variants cause the phenotype either by gain-of-function, or by dominant negative effects. The evidence for negative dominant is convincing for the Ile678Leu variant. However, the experiments and the discussion for the other two missense variants are not conclusive. Actually, the discussion is long and confusing and needs to be drastically reduced. Biochemical experiments are needed (particularly if KIF21B forms a dimer) to distinguish between the GOF and DomNeg alternatives.
3. The discussion could be reduced to half, without compromising its content.

More minor points

1. Some pathogenicity prediction programs provide non-pathogenic prediction for Gln313Lys and Ala1001Thr variants. Please discuss the comparison of this with your experimental functional data.
2. Please provide NM numbers of the transcript in the nomenclature of the variants in the different patients.

We thank the reviewers for their advises and suggestions. We provide below a detailed answer to the reviewers' concerns. As we have added a substantial amount of data, we have added a figure and a related supplemental figure (Figure 4 and supplementary figure 4 in the revised manuscript). We hope that, with these substantial revisions, our manuscript will fit with the reviewers' requirements and be suitable for publication in *Nature Communications*.

Reviewers' comments:

Reviewer #1 (Remarks to the Author):

In this paper the authors have identified several KIF21B amino-acid change mutations from human intellectual/neurodevelopmental disability patients with partial changes in MRI, and then characterized their way of dysfunction using in vitro and in vivo model systems of neuronal migration and axonal outgrowth. Both of the motor domain mutation and coiled-coil domain mutation gave out significant phenotypes, but only the former one could be rescued by wild-type cDNA overexpression. The reason of this discrepancy was speculated that the latter one would impair the autoinhibition so that it might be a gain-of-function mutant.

As pointed by the reviewer, all the three missense KIF21B variants (one in the motor domain (p.Gln313Lys), one in the coiled coil domain (p.Ile678Leu) and one in the regulatory coiled coil domain (p.Ala1001Thr)) are leading to migration phenotypes. We demonstrated that expression of all variants impairs neuronal migration (revised **Figure 3a,b**). However, it is worth clarifying here that expression of wild type KIF21B failed to rescue the migration phenotype induced by the 3 missense variants (revised **Figure 3a,b**), suggesting that **all KIF21B variants** (and not only the p.Ile678Leu variant, as mentioned above) unlikely have dominant negative effect (in the context of migration). Therefore, in migrating neurons, we suggest that **all the variants** rather act through a gain of function mechanism via attenuation of kinesin autoinhibition. In addition, we show that the ACC-related p.Ile678Leu variant perturbs axonal growth (revised **Figure 6a,b,d,e**) and ipsilateral axon branching (revised **Figure 7a-d**) through gain-of-function and dominant negative mechanisms respectively. In these last experiments, the p.Gln313Lys variant has been used as a negative control as patient do not show any commissural defects.

The following table is summarizing these findings.

	Migration	Axonal branching	Axonal growth
p.Gln313Lys	GoF*	no phenotype	no phenotype
p.Ile678Leu	GoF*	GoF*	DN**
p.Ala1001Thr	GoF*	no phenotype [#]	no phenotype [#]

* not rescued by overexpression of WT cDNA

** rescued by overexpression of WT cDNA

tested *in vitro*

in red : GoF : attenuation of autoinhibition that leads to enhanced kinesin motility activity (p.Ile678Leu- Δ ATPhKIF21B)

in orange : hypermobility has now been tested and the results have been added in the revised **Figure 4i,j** (p.Gln313Lys- Δ ATPhKIF21B, p.Ala1001Thr- Δ ATPhKIF21B).

GoF = Gain of function; DN= dominant negative.

The confusion might come from the way our findings are discussed (also pointed by R2). In the revised manuscript, the discussion has been substantially reduced and simplified to facilitate the reading and to avoid any confusion.

This manuscript will merit for publication because it may provide one of the first experimental evidence that KIF21B mutation is responsible for human neurodevelopmental abnormality in the CNS.

We thank the reviewer for highlighting the novelty of our study.

The weak points are [1] that the cell biological investigation on the causal relationship (such as the cargo identification and motor activity assay) is insufficient and [2] that the intellectual/behavioral abnormality was not assessed using the genetic models. Accordingly, this reviewer will request the authors to improve it along the direction of [1] and [2], to meet the quality of this class of journals.

[1] We demonstrate that, in contrast to the p.Ile678Leu mutant KIF21B that induces a severe migration defect, the expression of a p.Ile678Leu mutant protein that cannot bind to ATP induces a mild phenotype (revised **Figure 3i,j**; revised **Figure 6a,b,e**). We therefore suggest that the mutant protein exerts its gain of function effect by enhancing KIF21B motility activity. We agree with the reviewer that investigation of an effect of the variants on the mobility would bring direct evidences and strengthen our conclusion. For this reason, we have performed further investigations and clarified this aspect of our study. Experiments are detailed below (specific point 1).

[2] The reviewer proposes to assess whether the *KIF21B* variants are involved in the cognition phenotype observed in patient (Intellectual disability being a common clinical feature of the patients-**Table 1**). Though we agree that this is an important issue to study, here, we provide a clear demonstration of a causal relationship between variants in *KIF21B* and neurodevelopmental anomalies. In particular, we showed that expression of *KIF21B* variants specifically recapitulates patients' neurodevelopmental abnormalities. As such, the expression of the microcephaly-related variant reduces head size (revised **Figure 5**) but does not alter the formation of the cortical commissure (revised **Figures 6 and 7**). Conversely, the expression of the ACC-related variant impedes cortical intra- and inter-hemispheric connectivity (revised **Figures 6 and 7**) without having any effect on brain size (revised **Supplementary Figure 5c,d**). We think that these data are strong indicators to propose *KIF21B* as a major locus for neurodevelopmental disorders. We acknowledge that, in addition to our clinical data, more information would be required to definitely conclude about a causal relationship between *KIF21B* and cognitive impairment. We think this would be a study on his own and propose to moderate our conclusion in the revised version of the manuscript. We therefore changed the title to: "Mutations in the KIF21B kinesin gene cause neurodevelopmental disorders through imbalanced canonical motor activity".

Of note, Patient 3 has been evaluated for IQ during the revision process. These tests have demonstrated a **mild to moderate** intellectual disability with an IQ of 54-59 (verbal, performance).

Specific points:

1. The authors should assess the possible changes in microtubule-based motor ATPase activity of wild type and mutant KIF21B proteins to see whether the autoinhibition is really affected. The easiest assay to accomplish this will be transfecting each GFP-tagged construct to fibroblasts to see the difference in distribution/motility (Zhao et al., Cell 2001).

To address this comment, we tested the effect of the variants on KIF21B motility as follows:

1/ We assessed the distribution of HA-tagged wild type and mutant KIF21B proteins in neuronal ST cells. While WT, p.Gln313Lys and p.Ala1001Thr KIF21B proteins showed similar diffuse cytoplasmic localization, the p.Ile678Leu KIF21B protein tended to form aggregates localized mainly at the periphery of the cells, suggesting an enhanced motility toward the plus end of the microtubules. Those new data are presented as revised **Figure 4e**.

2/ We compared the processivity of the GFP-tagged wild type and mutant KIF21B proteins *in vitro* in Cos7 cells using videomicroscopy. This new set of experiments (see revised **Figure 4g and 4h**) revealed

a shift of GFP-KIF21B velocity toward high speed for all variant proteins compared to the WT protein. In addition, we observed an increase average velocity of the p.Ile678Leu variant compared to the wild-type protein.

Altogether these data strengthen the conclusion that the variants enhanced KIF21B processive activity. Noteworthy, the increase in KIF21B processivity correlated with the extent of migration defects, the velocity of p.Ile678Leu variant being more drastically enhanced. This reinforces the possibility that the degree of autoinhibition release imposed by the different missense variants dictates the severity of the phenotype as mentioned in the discussion section.

2. What is the most responsible binding partners of KIF21B for these phenotypes? The speculations in the Discussion section should be actually tested by cell biological analyses.

As mentioned in the introduction of our manuscript, very few is known about the cargos of KIF21B. We estimate that identifying new cargos of KIF21B through mass spectrometry analysis is out of the scope of our study and would not necessary help understanding the dysfunction of KIF21B variants since we consider that mutant proteins might favor the transport of cargos that are not transported by the wild type protein.

We agree that we haven't tested the hypothesis raised in the discussion. In the revised version of our manuscript, we have now tested the effect of the variants on the trafficking of potential responsible cargoes, BDNF/TrkB vesicles and mitochondria. We assessed the dynamics of BDNF-containing vesicles (BDNF-mCherry) as well as mitochondria (mito-RFP) in Cos7 cells overexpressing WT or mutant KIF21B by fast videomicroscopy. The velocity of BDNF vesicles was not increased in cells expressing WT or mutant KIF21B. Similarly, the velocity and the percentage of motile mitochondria remained unchanged in cells expressing the variant proteins. Therefore, excessive trafficking of BDNF vesicle or mitochondria is unlikely to be responsible of the phenotypes induced by mutant KIF21B. This further suggests that KIF21B variants might enhance motility of other unidentified cargoes. These results are now provided as **Supplementary Figure 4e-i**.

3. Do these mutant constructs also affect the cognitive flexibility in behavioral tests? It might be best to generate the mutant model mice with CRISPR/Cas9 technology, but a less time-consuming way of testing this will be injecting AAV-mediated expression vectors directly into hippocampus to see whether contextual fear extinction is affected (Morikawa et al., Cell Rep 2018).

As stated above, we believe that the experiments proposed by the reviewer might be a project on its own. Indeed, it would require the generation of a knock-in model. Getting the first validated mice and the cohort for behavioral analysis would take about one year. In addition to those analysis, further work would be needed to fully validate the model (neuroanatomy, protein expression and localization). We therefore estimate the minimum time required to answer properly to the reviewer's comment at 18 months. To us, this is not realistic.

The alternative experiment proposed by the reviewer involves injection of AAV in the hippocampus and analysis of contextual fear extinction. First, although we understand the point of injecting AAV in the hippocampus to assess the contextual fear extinction, we think that neither the structure, nor the tests are the most appropriate for our study. To us, it would be more accurate to target the cortex and perform classical cognitive tests (social discrimination (novel vs. familiar congener), interaction, recognition (congener vs. object), ultrasound vocalization, learning and memory (e.g. novel object recognition, Morris water maze), anxiety (e.g. elevate water maze)). Those tests are recommended by the International Phenotyping Mouse Consortium (IMPC) and widely used for analysis of cognition by the Gencodys consortium (Genetic and Epigenetic Networks in Cognitive Dysfunction). Second, the cognitive deficit described in the patient might be a consequence of neurodevelopmental anomalies, it does not seem appropriate to inject AAV after birth, in pups or young adult. For these reasons, we don't think the alternative experiments suggested by the reviewer would help assessing the cognitive defects.

4. Fig. 1e–g. The methods of weighted imaging in the MRI study appeared different from patient to patient. This should be unified for comparison.

We now provide an updated **Figure 1** with comparable T1 MRI sequence for all the patients.

Reviewer #2 (Remarks to the Author):

In this manuscript Asselin et al provide initial evidence that likely pathogenic variants in the KIF21B gene could cause a cognitive phenotype and brain abnormalities.

Significant Criticisms:

1. The evidence that the frameshift *de novo* variant in patient 4 is causative is not at all convincing. KIF21B has a PLI of 0 and therefore haploinsufficiency is likely to be tolerated. There are numerous truncating alleles in gnomAD and BRAVO apparently not related to a cognitive phenotype. In addition, the Kif21B1 +/- KO heterozygotes mice do not show a phenotype. This reviewer strongly suggests to delete patient 4 and the part of the paper related to this case.

We agree with the reviewer that, as discussed in our manuscript, the pathogenicity of the frameshift *de novo* variant is questionable. In the revised version of our manuscript, we have adjusted the text and indicated more clearly that this variant is possibly not pathogenic (see discussion page 14).

2. Since the PLI of KIF21B is 0, the expectation is that the non-synonymous variants cause the phenotype either by gain-of-function, or by dominant negative effects. The evidence for negative dominant is convincing for the Ile678Leu variant. However, the experiments and the discussion for the other two missense variants are not conclusive. Actually, the discussion is long and confusing and needs to be drastically reduced. Biochemical experiments are needed (particularly if KIF21B forms a dimer) to distinguish between the GOF and DomNeg alternatives.

We demonstrated that expression of all variants impairs neuronal migration (revised **Figure 3a,b**). Expression of wild type KIF21B failed to rescue the migration phenotype induced by the 3 missense variants (revised **Figure 3a,b**), suggesting that **all KIF21B variants unlikely affect migration through a dominant negative mechanism**. Therefore, in migrating neurons, we suggest that **all the variants** rather act through a gain of function mechanisms via attenuation of kinesin autoinhibition that leads to enhanced kinesin motility activity. To test this possibility, we analyzed the effect of expressing **immotile KIF21B mutant proteins** on neuronal migration. We originally presented the results for the p.Ile678Leu hKIF21B that lacks the ATP binding domain (p.Ile678Leu-ΔATPhKIF21B) and demonstrated that preventing the motility of the p.Ile678Leu mutant protein decreases the severity of the migration phenotype (revised **Figure 4i,j**). In the revised manuscript, we now have analyzed the impact on migration of the two other hKIF21B variants, p. Ala1001Thr and p.Gln3131Lys, that cannot bind ATP (NeuroD-p.Ala1001Thr-ΔATPhKIF21B; NeuroD-p.Gln3131Lys-ΔATPhKIF21B). Neurons expressing those immotile p.Ala1001Thr and p.Gln3131Lys variants showed a correct distribution, reinforcing our conclusion that, in migrating neuron, all the variants act through enhanced processive activity (revised **Figure 4i-k**).

We further rule out a dominant negative effect of the p. Ala1001Thr and p.Gln3131Lys variants: we performed two complementary assays both demonstrating that only the p.Ile678Leu protein might act as a dominant negative protein: **1.** We assessed the localization of the WT KIF21B protein in Cos7 cells expressing the KIF21B variants. The p.Ile678Leu variant but not the p. Ala1001Thr or the p.Gln3131Lys variant impaired the localization of the WT protein (revised **Figure 4f**). **2.** We performed immunoprecipitation experiments to test the ability of wild type protein to dimerize in presence of the I678L variant. Our results showed that the p.Ile678Leu variant is competing with the wild type protein to form KIF21B homodimer (revised **Supplementary Fig. 4d**).

3. The discussion could be reduced to half, without compromising its content.

As suggested, the discussion has been largely reduced in the revised version of our manuscript.

More minor points

1. Some pathogenicity prediction programs provide non-pathogenic prediction for Gln313Lys and Ala1001Thr variants. Please discuss the comparison of this with your experimental functional data.

In silico prediction tools such as SIFT, Polyphen-2 and others are useful to give insights into the pathogenicity of new missense variants but none of these tools can accurately predict whether a variant is pathogenic or not and have to be used with caution. The three missense variants described in our study all have concordant damaging predictions for SIFT and Polyphen-2 which are the two main prediction tools used in clinical routine and they also have a CADD score above 25, which is the threshold used for possibly pathogenic variants. The fact that other tools give different predictions is not surprising, as this is also the case for many class 5 missense variants in different genes. We believe that our functional study is much more useful to predict pathogenicity than any in silico algorithm.

2. Please provide NM numbers of the transcript in the nomenclature of the variants in the different patients.

NM numbers are now provided in the nomenclature of the variants page 5.

REVIEWERS' COMMENTS:

Reviewer #1 (Remarks to the Author):

In this revision the authors have improved their manuscript along the previous reviewers' suggestions. Although the fundamental conceptual advance in cell biology was not achieved, the authors have succeeded in obtaining data explaining the pathogenicity of the kinesin motor mutations that will merit for the general readers. However, new fig 4f lacks the test for reproducibility, so please statistically examine the change in kinesin accumulation. Accordingly, this reviewer will be glad to recommend its publication after improving this minor point.

Reviewer #2 (Remarks to the Author):

I recommend publication without the inclusion of patient 4, since the frameshift in this case is not the cause of the phenotype.

We thank the reviewers for recommending publication of our manuscript in *Nature Communications*. We provide below a detailed answer to the last reviewers' concerns.

REVIEWERS' COMMENTS:

Reviewer #1 (Remarks to the Author):

In this revision the authors have improved their manuscript along the previous reviewers' suggestions. Although the fundamental conceptual advance in cell biology was not achieved, the authors have succeeded in obtaining data explaining the pathogenicity of the kinesin motor mutations that will merit for the general readers.

We thank the reviewer for acknowledging our effort to better explain the pathogenicity of the identified variants.

However, new fig 4f lacks the test for reproducibility, so please statistically examine the change in kinesin accumulation. Accordingly, this reviewer will be glad to recommend its publication after improving this minor point.

In this experiment, ST cells were transfected with WT- hKIF21B Myc-tagged construct and WT or mutant HA-tagged hKIF21B constructs. 100% of the ST cells showing a mislocalization of the p.Ile678Leu variant, display altered cellular localization of the Myc-tagged WT protein. Notably, none of the cells expressing the WT or other variants HA-KIF21B, show impaired localization of the Myc-tagged WT protein. This has been clarified in the revised manuscript (main text, page 9).

Reviewer #2 (Remarks to the Author):

I recommend publication without the inclusion of patient 4, since the frameshift in this case is not the cause of the phenotype.

As suggested by the editor, we kept the data associated to patient 4.